# Humans actively sample evidence to support prior beliefs

Paula Kaanders[1,2]*, Pradyumna Sepulveda[3], Tomas Folke[4,5], Pietro Ortoleva[6], Benedetto De Martino[3,7]*

[1]Department of Experimental Psychology, University of Oxford, Oxford, United Kingdom; [2]Wellcome Centre for Integrative Neuroimaging, University of Oxford, Oxford, United Kingdom; [3]Institute of Cognitive Neuroscience, University College London, London, United Kingdom; [4]Department of Mathematics and Computer Science, Rutgers University, Newark, United States; [5]Centre for Business Research, Cambridge Judge Business School, University of Cambridge, Cambridge, United Kingdom; [6]Department of Economics and Woodrow Wilson School, Princeton University, Princeton, United States; [7]Wellcome Centre for Human Neuroimaging, University College London, London, United Kingdom

*For correspondence:
paula.kaanders@psy.ox.ac.uk (PK);
benedettodemartino@gmail.com (BDM)

Competing interest: The authors declare that no competing interests exist.

**Abstract** No one likes to be wrong. Previous research has shown that participants may underweight information incompatible with previous choices, a phenomenon called confirmation bias. In this paper, we argue that a similar bias exists in the way information is actively sought. We investigate how choice influences information gathering using a perceptual choice task and find that participants sample more information from a previously chosen alternative. Furthermore, the higher the confidence in the initial choice, the more biased information sampling becomes. As a consequence, when faced with the possibility of revising an earlier decision, participants are more likely to stick with their original choice, even when incorrect. Critically, we show that agency controls this phenomenon. The effect disappears in a fixed sampling condition where presentation of evidence is controlled by the experimenter, suggesting that the way in which confirmatory evidence is acquired critically impacts the decision process. These results suggest active information acquisition plays a critical role in the propagation of strongly held beliefs over time.

## Editor's evaluation

Kaanders et al. investigate how the sampling of visual information by human subjects is biased toward their previous choice. The novel experiments and rigorous analyses largely support the presence of a "confirmation bias" that arises specifically when information sampling is under subjects' control. These findings should be of interest to a broad community ranging from decision-making to metacognition research.

## Introduction

We are constantly deciding what information to sample to guide future decisions. This is no easy feat, as there is a vast amount of information available in the world. In theory, an agent should look for information to maximally reduce their uncertainty (*Schulz and Gershman, 2019*; *Schwartenbeck et al., 2019*; *Yang et al., 2016*). There is some evidence that this is indeed what humans do (*Kobayashi and Hsu, 2019*; *Meyer and Shi, 1995*; *Steyvers et al., 2009*; *Wilson et al., 2014*), but a range of other drivers of information search have been identified that make us deviate from the sampling behaviour of optimal agents (*Eliaz and Schotter, 2007*; *Gesiarz et al., 2019*; *Hunt et al., 2016*; *Kobayashi*

*et al., 2019*; *Rodriguez Cabrero et al., 2019*; *Sharot, 2011*; *van Lieshout et al., 2018*; *Wang and Hayden, 2019*).

Confirmation bias is defined as the tendency of agents to seek out or overweight evidence that aligns with their beliefs while avoiding or underweighting evidence that contradicts them (*Hart et al., 2009*; *Lord et al., 1979*; *Nickerson, 1998*; *Stroud, 2007*; *Wason, 1960*; *Wason, 1968*). This effect has often been described in the context of what media people choose to consume (*Bakshy et al., 2015*; *Bennett and Iyengar, 2008*; *Pariser, 2012*). Recently, cognitive scientists have realised confirmation bias may not be restricted to this domain and may reveal a fundamental property of the way in which the brain drives information search. As such, researchers have started to investigate confirmation bias in simple perceptual and value-based choice tasks. For example, sensitivity to new information has been shown to decrease after choice (*Bronfman et al., 2015*), while *Talluri et al., 2018* have shown that sensitivity to a stimulus is increased when it is consistent with the stimulus presented before. Furthermore, in *Palminteri et al., 2017a* learning rates were found to be higher for feedback indicating positive obtained outcomes as well as for negative forgone outcomes. This effect has been replicated in recent studies that suggest confirmation bias during learning might be adaptive in some contexts (*Tarantola et al., 2021*; *Lefebvre et al., 2020*; *Salem-Garica et al., 2021*).

Most of the existing research on perceptual decision-making has focused on the weighting of incoming information during evidence accumulation or belief updating. Other findings suggest that agents also actively search for confirmatory evidence (*Hunt et al., 2016*; *Hunt et al., 2018*; *Tolcott et al., 1989*; *Wason, 1968*), but none of these experiments investigated the effect of biased information sampling in simple, perceptual choice. As most paradigms currently used to study confirmation bias do not explicitly focus on active information sampling performed by the agent, therefore it is unclear whether confirmation bias arises from evidence underweighting, biased information sampling, or both.

Another important aspect of confirmation bias is how it modulates and is modulated by decision confidence. Confidence is known to play an important role in guiding decisions (*Bogacz et al., 2010*; *Boldt et al., 2019*; *Folke et al., 2016*; *Rabbitt, 1966*; *Yeung and Summerfield, 2012*). It is therefore likely it may also guide the information search preceding choice (*Desender et al., 2019*). In fact, *Rollwage et al., 2020* recently showed that confidence indeed affects a neural signal of confirmatory evidence integration. By definition, the higher confidence is in a choice, the stronger the decision-maker's belief is in the correctness of this choice (*Fleming and Lau, 2014*; *Pouget et al., 2016*). Accordingly, we predict that confirmation bias in information sampling will be stronger after choices made with high confidence.

We used two perceptual binary forced choice tasks with two choice phases separated by a free sampling phase to test our hypotheses that confirmation bias arises from biased information sampling, that this effect influences future choice and that confidence mediates this behavioural tendency.

## Results

Here, we present the results of two independent experiments we conducted to test our hypotheses.

### Experiment 1

Participants performed a perceptual two-alternative forced choice (2AFC) task (*Figure 1A*), in which they were briefly presented with patches containing multiple dots. Participants were asked to judge which patch contained the most dots. After their initial choice (phase 1), they rated their confidence that they were correct. Subsequently, in the second phase of the trial (phase 2) participants could freely sample (i.e., see) each dot patch using key presses to switch between them as frequently as they liked for 4000ms. Participants could only view one patch at a time. They were then prompted to make a second choice (phase 3), in which they could either confirm their initial choice or change their mind. After this second decision, they again reported their degree of confidence.

As expected, participants' performance was highly sensitive to the amount of perceptual evidence presented in a given trial (i.e. the difference in the amount of dots in the left and right patches; *Figure 1B*). Participants were also more accurate in the second choice compared to the first choice, showing that they used the additional sampling time between choice phases to accumulate additional

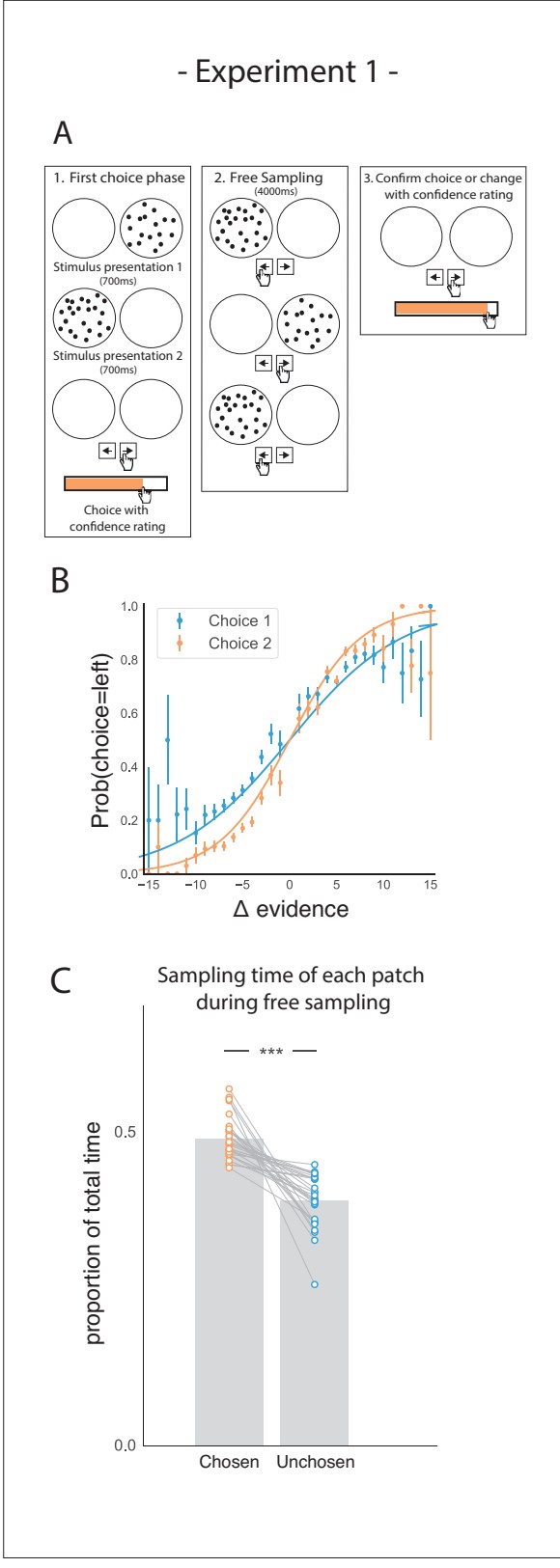

**Figure 1.** Task design and participant behaviour for experiment 1. (**A**) Task structure. Participants had to choose which of two dot patches contained the most dots after viewing each for 700ms (phase 1) and rate their confidence in this choice. Then participants were given 4000ms to view the dots which they could allocate between the two patches in whichever way they liked (phase 2) by pressing the left and right arrow keys. Finally, participants made a

*Figure 1 continued on next page*

*Figure 1 continued*

second choice about the same set of stimuli and rated their confidence again (phase 3). (**B**) Participants effectively used the stimuli to make correct choices and improved upon their performance on the second choice. This psychometric curve is plotting the probability of choosing the left option as a function of the evidence difference between the two stimuli for each of the two choice phases. (**C**) In the free sampling phase (phase 2) participants spent more time viewing the stimulus they chose on the preceding choice phase than the unchosen option. Data points represent individual participants.

perceptual evidence that allowed them to increase accuracy on the second choice ($t_{27}$ = 8.74, p < 0.001).

Our main hypothesis was that participants would prefer to gather perceptual information that was likely to confirm their initial decision. Conversely, we expected them to be less likely to acquire evidence that would disconfirm it. To perform well on this task, an agent would have to attend equally to the two patches, as the goal requires computing the difference in dots between the two patches. Therefore, any imbalance in sampling time would not necessarily be beneficial to completing the task. However, we expected that in the free sampling phase (phase 2) of the trial participants would spend more time viewing their chosen patch. In line with this hypothesis, during the sampling phase, participants spent more time viewing the patch they chose in the first decision phase relative to the unchosen alternative (*Figure 1C*; $t_{27}$ = 7.28, p < 0.001).

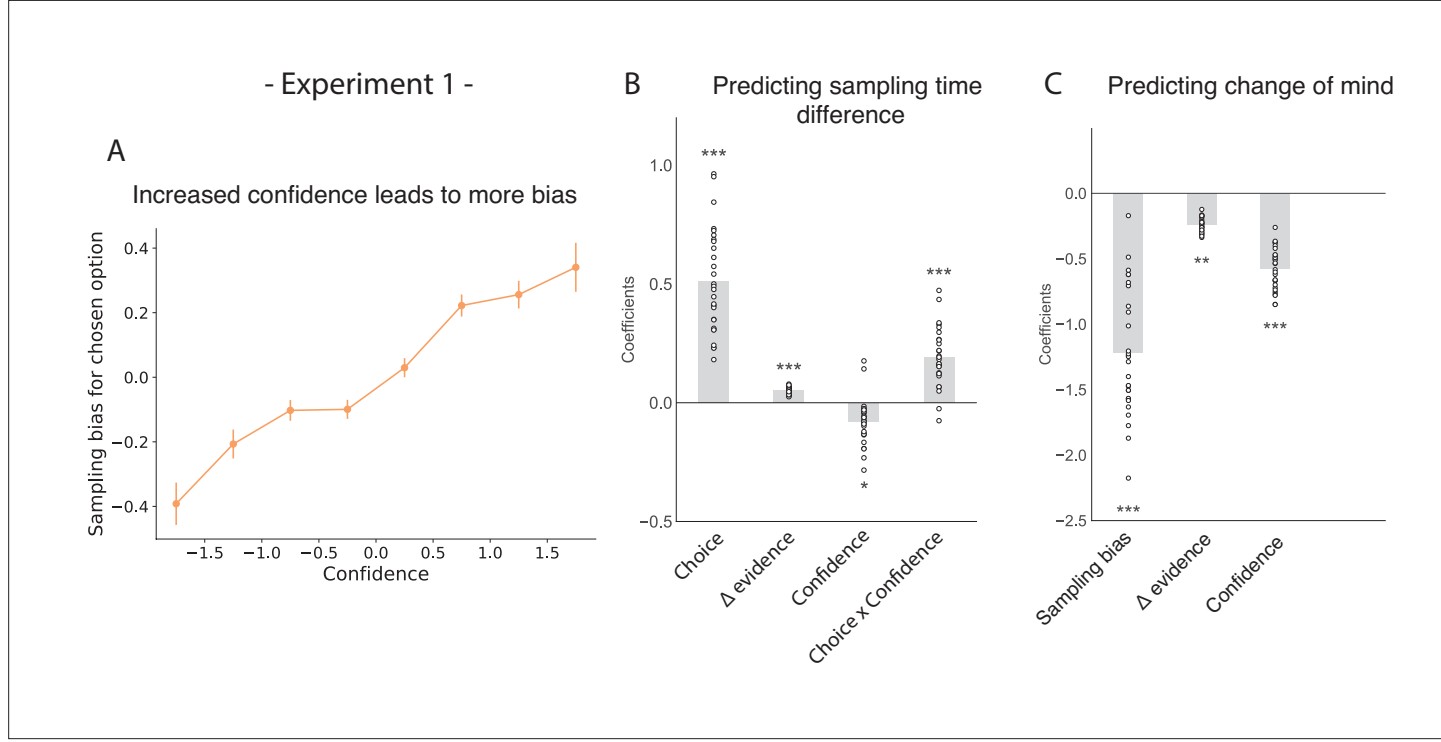

**Figure 2.** The effect of choice on sampling behavior is mediated by confidence in experiment 1. Participants were less likely to change their mind if they showed a strong sampling bias for their initially chosen option in the sampling phase. (**A**) Sampling bias in favour of the chosen option increases as a function of confidence in the initial choice. Confidence and sampling bias are both normalised at the participant level in this plot. (**B**) There is a significant main effect of choice on sampling time difference, such that an option is sampled for longer if it was chosen, and a significant interaction effect of Choice x Confidence, such that options chosen with high confidence are sampled for even longer. (**C**) There is a main negative effect of sampling bias on change of mind, such that participants were less likely to change their mind in the second decision phase (phase 3) the more they sampled their initially chosen option in the free sampling phase (phase 2). (**B–C**) Plotted are fixed-effect coefficients from hierarchical regression models predicting the sampling time (how long each patch was viewed in the sampling phase) difference between the left and right stimuli. Data points represent regression coefficients for each individual participant.

The online version of this article includes the following figure supplement(s) for figure 2:

**Figure supplement 1.** Confidence in the second choice is significantly predicted by trial difficulty (dot difference) and change of mind.

Confidence in a choice reflects the strength of the participant's belief that their choice was correct. Consequently, we hypothesised that participants' preference for gathering confirmatory evidence for a given choice would be modulated by their confidence in that choice. As such, we expected choices made with higher confidence would lead to a stronger sampling bias favouring the chosen patch over the unchosen patch. In a hierarchical regression predicting sampling time difference between the chosen and unchosen stimulus, we found a significant interaction between choice and confidence, such that the higher the degree of confidence was in the first choice, the more sampling was biased in favour of that choice (*Figure 2*; $t_{26.96}$=5.26, p < 0.001; see Appendix 1). Besides this, sampling time difference in turn affected confidence in the second choice, such that the stronger sampling bias was towards the initially chosen stimulus, the higher confidence in the second choice if that same stimulus was chosen again (*Figure 2—figure supplement 1*; $t_{26.01}$=9.40, p < 0.001). Change of mind was a significant negative predictor of the second confidence rating (*Figure 2—figure supplement 1*; $t_{23.66}$=-10.41, p < 0.001). Also, a negative interaction effect between sampling time difference and change of mind on the second confidence rating was observed, meaning that the effect of sampling bias on confidence was reversed in trials where participants changed their mind (*Figure 2—figure supplement 1*; $t_{24.48}$=-10.21, p < 0.001).

We also saw a significant main positive effect of choice on sampling time difference, such that a chosen stimulus was likely to be sampled for longer during the sampling phase (*Figure 2B*; $t_{26.96}$=9.64, p < 0.001) as shown in the previous section. There was also a significant main positive effect of evidence difference on sampling time difference, whereby the correct stimulus (containing the most dots) was sampled for longer (*Figure 2B*; $t_{26.96}$=13.02, p < 0.001).

Our choices are determined by the evidence we have gathered before making a decision. Therefore, we expected that biased sampling in the free sampling phase (phase 2) would impact decisions in the second decision phase (phase 3). Specifically, we hypothesised that the more strongly participants preferred sampling their chosen patch, the more likely they were to choose it again. Using a multinomial hierarchical logistic regression, we found that bias in sampling time towards the previously chosen option was indeed predictive of future choice. In other words, the more participants sampled a previously chosen option, the more likely they were to choose it again (*Figure 2C*; z = −11.0, p < 0.001; see Appendix 2). Furthermore, evidence difference and confidence were significantly negatively related to subsequent changes of mind, whereby participants were less likely to change their mind if their initial choice was correct and if it was made with high confidence (*Figure 2C*; main effect of evidence difference – z = −3.06, p < 0.01, main effect of confidence – z = −10.12, p < 0.001).

## Experiment 2

While the results from the first experiment showed an effect of biased sampling on subsequent choice, it was not clear whether this effect arises from biased evidence accumulation caused by differential exposure to the perceptual evidence, or if the sampling choices themselves drive the effect. In other words, would the same choice bias appear if participants were passive recipients of biased sampling, or does the choice bias require that participants make their own sampling decisions?

We addressed this question in a follow-up eye-tracking study (*Figure 3A*) in which we introduced a control task, a 'fixed-viewing condition', in which participants did the same task, but did not have the possibility to freely sample the patches in phase 2. Instead, the dot patches were shown for a set amount of time. In one-third of trials, the patches were shown an equal amount of time; in two-thirds of trials, one patch was shown three times longer than the other. Each participant completed two sessions, one session involved free sampling, similar to Experiment 1. The other involved a fixed-viewing control task. Session order was pseudo-randomized between participants. Furthermore, presentation of the visual stimuli and all the participants' responses were gaze-contingent. This meant that during the sampling phase (phase 2), the dot patches were only presented when participants fixated inside the patch. Furthermore, we hypothesised that sampling bias might be stronger when more time to sample is available. Specifically, we expected that participants might become more biased in their sampling throughout the sampling phase as they gradually become more confident in their belief. Therefore, we manipulated the length of phase 2 to be either 3000ms, 5000ms, or 7000ms. Again, participants were sensitive to the difficulty of the given trials (*Figure 3B*) and were more accurate on the second choice compared to the first choice ($t_{17}$ = 6.80, p < 0.001).

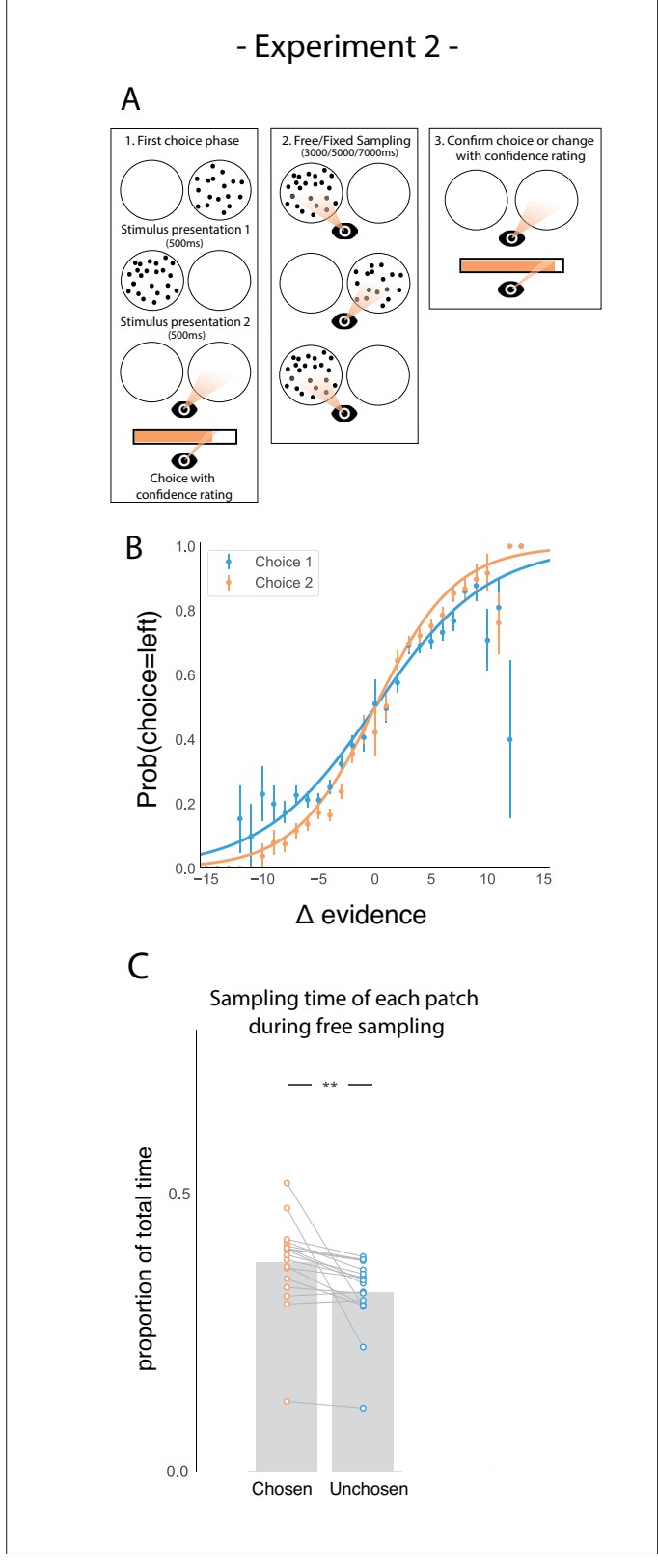

**Figure 3.** Task design and participant behaviour for experiment 2. (**A**) Task structure. Participants had to choose which of two dot patches contained the most dots after viewing each for 500ms (phase 1) and rate their confidence in this choice. Then participants were given 3000ms, 5000ms, or 7000ms to view the dots which they could allocate between the two patches in whichever way they liked (phase 2) by looking inside the circles. Finally, participants

*Figure 3 continued on next page*

*Figure 3 continued*

made a second choice about the same set of stimuli and rated their confidence again (phase 3). (**B**) Participants effectively used the stimuli to make correct choices and improved upon their performance on the second choice. This psychometric curve is plotting the probability of choosing the left option as a function of the evidence difference between the two stimuli for each of the two choice phases. (**C**) In the free sampling condition during the sampling phase (phase 2) participants spent more time viewing the stimulus they chose on the preceding choice phase than the unchosen option. Data points represent individual participants.

The online version of this article includes the following figure supplement(s) for figure 3:

**Figure supplement 1.** Mean sampling time viewing the initially chosen and unchosen patch in the sampling phase in study 2 for each sampling phase length and each condition.

We also replicated our main result in the free sampling condition showing that participants spent more time viewing the patch they just chose (*Figure 3C*; $t_{17}$ = 3.52, p < 0.01). Furthermore, the size of this sampling time bias was proportional to the total amount of sampling time available in study 2 (*Figure 3—figure supplement 1*), suggesting that there was no particular period of time within the sampling phase where confirmatory information sampling was more likely contrary to our expectation.

This new experiment also replicated the mediating effect of confidence on how sampling bias affects subsequent choice. We again found a significant interaction between choice and confidence (*Figure 4A–B*; $t_{16.97}$ = 4.29, p < 0.001; see supplemental materials Appendix 1) and replicated the main positive effects of choice and evidence difference on sampling time difference between the chosen and unchosen stimulus (*Figure 4B*; main effect of choice: $t_{16.97}$ = 2.90, p < 0.01; main effect of evidence difference: $t_{16.97}$= 9.21, p < 0.001). Confidence was also shown to negatively predict the total amount of time spent sampling, that is, the total time actually spent gazing at the two stimuli during the sampling phase (rather than the central fixation cross; *Figure 4—figure supplement 1*; $t_{22.99}$ = –4.01, p < 0.001). Change of mind also negatively predicted confidence in the second choice phase (*Figure 2—figure supplement 1*; $t_{16.17}$ = –6.16, p<0.001), and there was again a positive effect of sampling time difference on the second confidence rating ($t_{17.03}$ = 5.79, p<0.001) as well as a significant negative interaction effect between change of mind and sampling time difference ($t_{24.55}$ = –3.96, p<0.001).

Similarly, we replicated the negative effect of sampling bias on subsequent change of mind (*Figure 4C*; z = –7.20, p < 0.001; see Appendix 2) as well as the main negative effects of evidence difference and confidence on change of mind (*Figure 4C*; main effect of evidence difference: z = –2.66, p < 0.01; main effect of confidence: z = –8.73, p < 0.001).

It has been shown that the uncertainty around an internal estimate scales with numerosity (*Scott et al., 2015*). As such, an alternative explanation for the sampling biases found in both experiments might be that participants were minimizing uncertainty by sampling the option with more dots (the correct choice alternative) for longer. To rule out this possibility, we performed some of the regressions presented earlier in the Results section including numerosity (the total number of dots present on the screen) as a predictor (see *Figure 4—figure supplements 3–8*). We found that neither total numerosity (experiment 1: $t_{12.71}$=-1.77, p = 0.10; experiment 2: $t_{25.07}$=0.62, p = 0.54) or the number of dots of the chosen option (experiment 1: $t_{71.10}$=-1.82, p = 0.07; experiment 2: $t_{87.40}$=0.25, p = 0.81) had a significant effect on sampling bias, meaning that participants' sampling was not biased by a drive to reduce uncertainty by sampling the option with more dots. Furthermore, we reanalysed data from a previous perceptual decision study (*Sepulveda et al., 2020*) where participants also had to choose between two circle stimuli with dots (but were not requested to resample the stimuli and choose for a second time). In this study, on some trials participants had to choose the option with the most dots (the 'most' frame), and on other trials the option with the least dots (the 'fewest' frame). In this dataset, we found a significant effect of choice on sampling time difference in both frames ('most' frame: $t_{57.54}$=24.01, p < 0.001; 'fewest' frame: $t_{33.72}$=14.97, p < 0.001; *Figure 4—figure supplement 8*) and no significant effect of the total number of dots on sampling time difference ('most' frame: $t_{31.41}$=-0.37, p = 0.72; 'fewest' frame: $t_{40.01}$=1.49, p = 0.14), meaning that participants' sampling was not biased by numerosity. Overall, these results seem to indicate that numerosity is not significantly affecting the sampling process in this task.

The sequential order of presentation in the initial sampling phase before the first choice might also be expected to affect sampling. To exclude this possibility, we performed a regression analysis

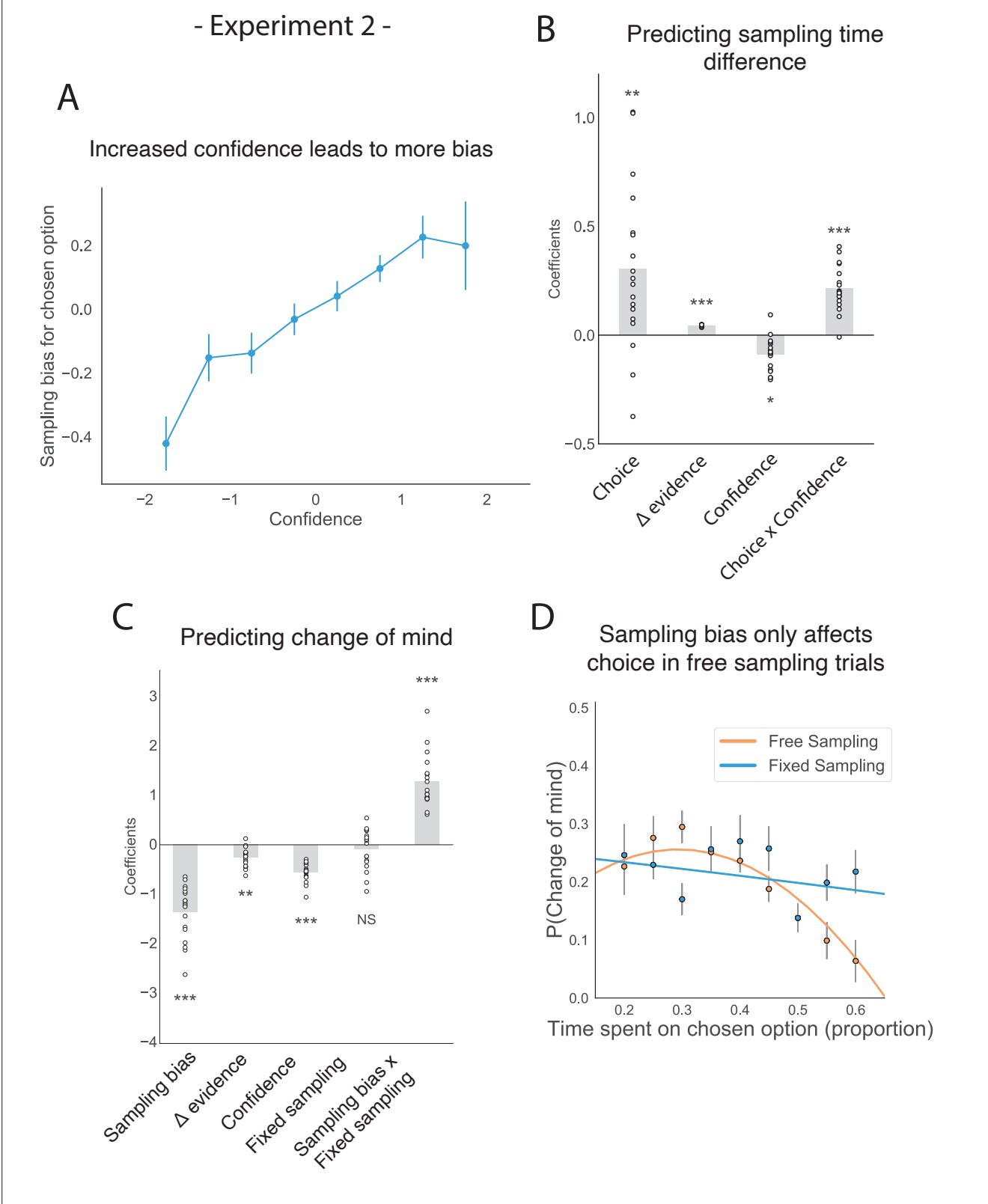

**Figure 4.** The effect of choice on sampling behaviour is mediated by confidence in experiment 2. Participants were less likely to change their mind if they showed a strong sampling bias for their initially chosen option in the sampling phase, but this was only the case in the free sampling condition. (**A**) Sampling bias in favour of the chosen option increases as a function of confidence in the initial choice. Confidence and sampling bias towards the chosen option are both normalized at the participant level in this plot. (**B**) There is a significant main effect of choice on sampling time difference, such

*Figure 4 continued on next page*

*Figure 4 continued*

that an option is sampled for longer if it was chosen, and a significant interaction effect of Choice x Confidence, such that options chosen with high confidence are sampled for even longer. (**C**) There is a main negative effect of sampling bias on change of mind, such that participants were less likely to change their mind in the second decision phase (phase 3) the more they sampled their initially chosen option in the free sampling phase (phase 2). The main effect of sampling bias on change of mind disappears in the fixed sampling condition, which can be seen by the positive interaction term Sampling bias x Fixed sampling which entirely offsets the main effect. The analysis includes a dummy variable 'Fixed Sampling' coding whether the trial was in the fixed-viewing condition. (**B–C**) Plotted are fixed-effect coefficients from hierarchical regression models predicting the sampling time (how long each patch was viewed in the sampling phase) difference between the left and right stimuli. Data points represent regression coefficients for each individual participant. (**D**) The probability that participants change their mind on the second choice phase is more likely if they looked more at the unchosen option during the sampling phase. The plot shows the probability that participants changed their mind as a function of the time spent sampling the initially chosen option during phase 2. The lines are polynomial fits to the data, while the data points indicate the frequency of changes of mind binned by sampling bias. Note that actual gaze time of the participants is plotted here for both task conditions. The same pattern can be seen when instead plotting the fixed presentation times of the stimuli for the fixed task condition (see *Figure 4—figure supplement 2*).

The online version of this article includes the following figure supplement(s) for figure 4:

**Figure supplement 1.** Confidence in the first choice reduces the total amount of time spent sampling (gazing at the two stimuli) in the free sampling trials in experiment 2.

**Figure supplement 2.** In *Figure 4D*, we plotted the probability of changes of mind as a function of actual gaze time by participants in the fixed viewing condition.

**Figure supplement 3.** Numerosity has no significant effect on sampling bias in a regression analysis predicting sampling bias with total numerosity (total number of dots present on a trial) included as a predictor.

**Figure supplement 4.** Numerosity has no significant effect on sampling bias in a regression analysis predicting sampling bias with numerosity of the chosen stimulus (dots in the chosen stimulus) included as a predictor.

**Figure supplement 5.** Numerosity had a small significant effect on accuracy in the first choice phase in experiment 1, such that participants made more mistakes on trials with high total numerosity (total number of dots).

**Figure supplement 6.** Confidence was not affected by numerosity in a linear regression model.

**Figure supplement 7.** Confidence change in experiment 1 was negatively affected by total numerosity (total number of dots), although it is a small effect.

**Figure supplement 8.** A sampling bias towards the stimulus that participants would end up choosing was found in an independent dataset from a perceptual experiment presented in *Sepulveda et al., 2020*, and this was not affected by total numerosity ()on a trial.

**Figure supplement 9.** Order of presentation has no significant effect on sampling bias in experiment 2.

**Figure supplement 10.** Choice behaviour in Experiment 2.

**Figure supplement 11.** There was no significant difference in the number of changes of mind from the incorrect to the correct option (**A**) or in the total number of changes of mind (**B**) between the free and fixed sampling conditions.

**Figure supplement 12.** Confidence ratings experiment 2.

predicting sampling time difference as a function of presentation order in experiment 2 and found no effect (*Figure 4—figure supplement 9*; $t_{49.65}=0.08$, p = 0.93). It is also important to note that the stimulus chosen in the first choice phase was highlighted throughout the trial. This was done to reduce the likelihood of working memory being a confound on this task, but we recognise the possibility that it may have interacted with the main effect of choice on sampling.

Once we confirmed all the main findings from the first experiment using this new setup, we were able to test our main hypothesis: does the effect of sampling bias on choice we identified require the participant to actively choose which information to sample? In other words, is the effect of confirmation bias on subsequent choice present when confirmatory evidence is passively presented to participants or does it require that confirmatory evidence is actively sought by the decision-maker? In the first case, we would expect to see the same effect in the fixed-viewing condition (in which asymmetric information is provided by the experimenter) as in the free-sampling condition. In the second case, we would expect that the effect of biased information sampling on the subsequent choice disappears in the fixed-viewing condition.

In line with the second prediction, we found that in the fixed-viewing condition, contrary to the free-sampling condition in this experiment and in the first experiment, the amount of time spent viewing the patches in the sampling phase did not significantly affect subsequent choice. In a multinomial hierarchical logistic regression predicting change of mind from the first choice phase to the second choice phase within a given trial, the main effect of sampling bias on change of mind was completely offset

by the positive effect of the interaction term between sampling bias and a dummy variable that was set to 1 if a trial was in the fixed-viewing condition (*Figure 4C–D*; z = 6.77, p < 0.001; see Appendix 2). This means that there was no effect of sampling bias on change of mind in the fixed-viewing condition. To check that participants were engaging in this version of the task, we looked whether the number of saccades made *within* each patch during the sampling phase was similar between the two tasks. We found that the number of saccades was actually higher in the fixed-viewing condition than in the main experiment ($t_{17}$ = −4.22, p < 0.001), which means participants were indeed exploring the information in this condition. Furthermore, no significant difference in accuracy was observed between the two conditions ($t_{17}$ = 1.51, p = 0.14), though sensitivity to decision evidence was slightly higher in the second choice in the free sampling condition compared to the fixed sampling condition. The number of changes of mind was also equal between the two conditions ($t_{17}$ = 0.75, p = 0.47) as well as both confidence ratings (confidence in first choice: $t_{17}$ = −1.38, p = 0.19; confidence in second choice: $t_{17}$ = 0.5, p = 0.62; for more details see *Figure 4—figure supplements 10–12*).

To further investigate how attention, when freely allocated, shapes the accumulation of evidence and choice bias, we modelled the data from both viewing conditions using the Gaze-weighted Linear Accumulator Model (GLAM; *Molter et al., 2019*; *Sepulveda et al., 2020*; *Thomas et al., 2019*). GLAM belongs to the family of race models with an additional modulation by visual attention (*Figure 5A*). It is an approximation of a widely used class of models – the attentional Drift Diffusion Model (aDDM; *Krajbich et al., 2010*; *Krajbich and Rangel, 2011*) in which the full dynamic sequence of fixations is replaced by the percentage of time spent fixating the choice alternatives. Even-numbered trials were used to fit the model while odd-numbered trials were used to test it. See the Materials and methods section for further details.

GLAM is defined by four free parameters: $\nu$ (drift term), γ (gaze bias), $\tau$ (evidence scaling), and σ (normally distributed noise standard deviation). The model correctly captured the reaction times (RT) and choice behaviour of the participants at group-level both in the free-sampling (*Figure 5B*) and fixed-viewing conditions (*Figure 5C*). More specifically, we found that the model predicted faster RTs when trial difficulty was low (|ΔDots| is high; *Figure 5B–C*, top left). The model also reproduced overall choice behaviour as a function of the number of dots in the patches (ΔDots = Dots$_{Left}$ − Dots$_{Right}$) in both conditions (*Figure 5B–C*, top right). Furthermore, we found gaze allocation (ΔGaze = g$_{Left}$ − g$_{Right}$) predicted the probability of choosing the correct patch in the free-sampling condition (*Figure 5C*, bottom left). However, to properly test how predictive gaze allocation is of choice, we must account for the effect of evidence (ΔDots) on choice. As such, we used the gaze influence (GI) measure (*Thomas et al., 2019*), which reflects the effect of gaze on choice after accounting for the effect of evidence on choice. GI is calculated taking the actual choice (0 or 1 for right or left choice, respectively) and subtracting the probability of choosing the left item as predicted by a logistic regression with ΔDots as a predictor estimated from behaviour. The averaged residual choice probability reflects GI. We found GI estimated purely from participant's behaviour was higher in the free-sampling than in the fixed-viewing condition (comparing average GI by participant, free-sampling condition: Mean = 0.148, SD = 0.169; fixed-viewing condition: Mean = 0.016, SD = 0.14; $t_{17}$ = 2.708, p < 0.05). This suggests the effect of visual attention on choice was higher in the free-sampling condition. In line with this, the model also predicted a higher GI on corrected choice probability in the free-sampling condition (comparing average GI by individual model predictions, free-sampling condition: Mean = 0.112, SD = 0.106; fixed-viewing condition: Mean = 0.033, SD = 0.028; $t_{17}$ = 2.853, p < 0.05; *Figure 5B–C*, bottom right).

We then tested whether attention affected information integration more when information was actively sought (i.e. the free-sampling condition) compared to when information was given to the participants (i.e. the fixed-viewing condition). We compared the parameters obtained from the individual fit in the free-sampling and fixed-viewing conditions (*Figure 5D*). We found a significant variation in the gaze bias parameter (Mean γ $_{Free}$ = 0.81, Mean γ $_{Fixed}$ = 0.98, $t_{17}$ = −3.934; p < 0.01), indicating a higher influence of gaze on choice in the free-sampling condition. Note that during the fixed-viewing condition, the parameter γ ≈ 1 indicates that almost no gaze bias was present on those trials. Conversely, there was no significant difference for the other parameters between the conditions (Mean $\tau$ $_{Free}$ = 1.44, $\tau$ $_{Fixed}$=1.13, $t_{17}$ = 1.003; p = 0.32, n.s.; Mean σ$_{Free}$ = 0.0077, Mean σ$_{Fixed}$=0.0076, $t_{17}$ = 0.140; p = 0.89, n.s.; $\nu$ $_{Free}$=8.07x10$^{-5}$, $\nu$ $_{Fixed}$=8.70x10$^{-5}$, $t_{17}$ = −1.201; p = 0.24, n.s.). These results suggest that gathering information actively (i.e. free-sampling condition) does not affect the overall

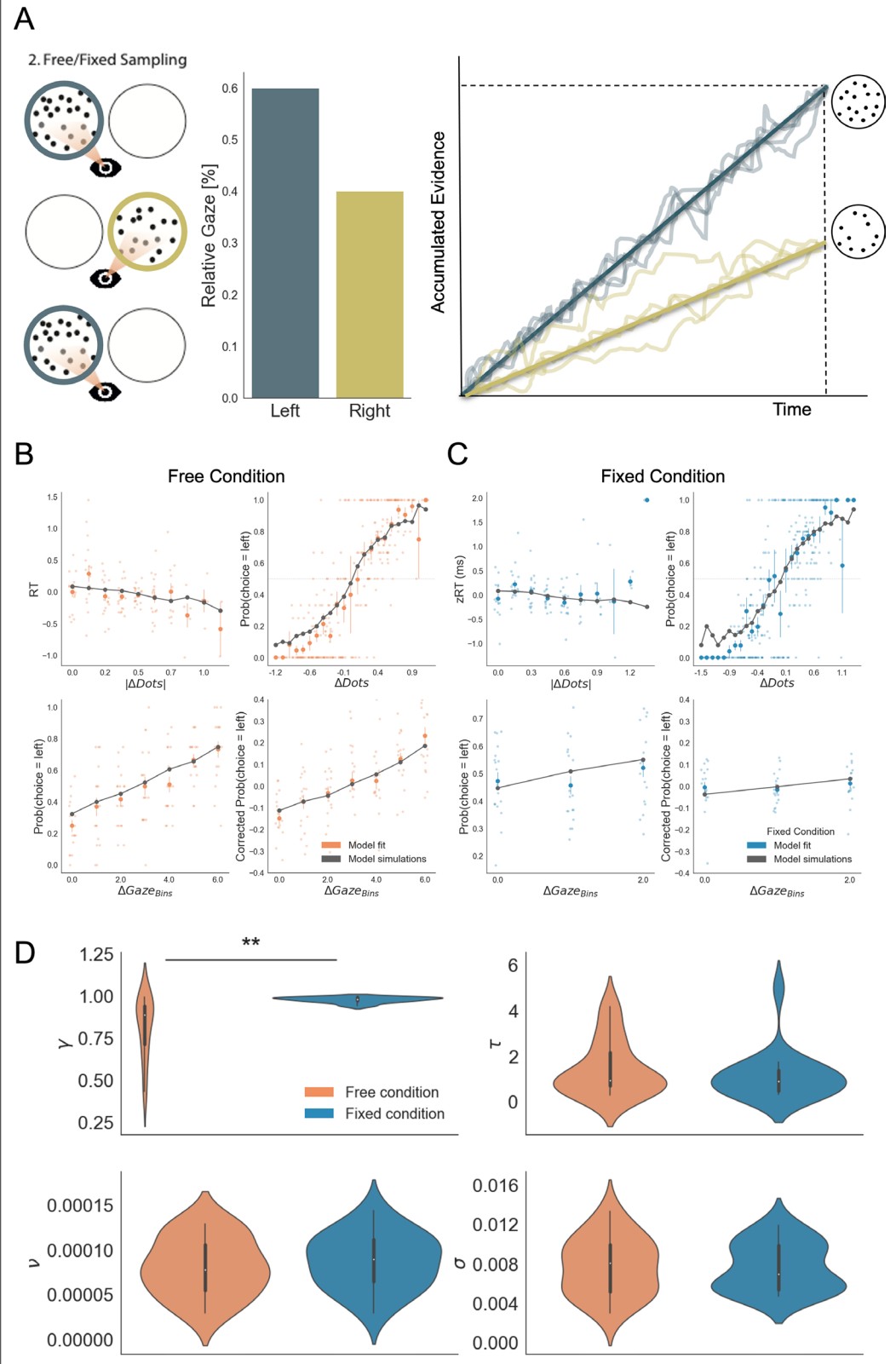

**Figure 5.** Gaze impacted evidence accumulation (for the 2<sup>nd</sup> choice) more strongly in the free than in the fixed sampling condition. (**A**) Free and fixed sampling condition trials were fitted separately using a Gaze-weighted Linear Accumulator Model (GLAM). In this model there are two independent accumulators for each option (left and right) and the decision is made once one of them reaches a threshold. Accumulation rate was modulated

*Figure 5 continued on next page*

*Figure 5 continued*

by gaze times, when gaze bias parameter is lower than 1 ($\gamma < 1$). In that case, the accumulation rate will be discounted depending on $\gamma$ and the relative gaze time to the items, within the trials. Gaze information from the free sampling trials, and presentation times from the fixed sampling trials were used to fit the models. The panel depicts an example trial: patch sampling during phase 2 (left panel) is used to estimate the relative gaze for that trial (central panel), and the resulting accumulation process (right panel) Notice GLAM ignores fixations dynamics and uses a constant accumulation term within trial (check *Methods* for further details). The model predicted the behaviour in free (**B**) and fixed (**C**) sampling conditions. The four panels present four relevant behavioural relationships comparing model predictions and overall participant behaviour: (top left) response time was faster (shorter RT) when the choice was easier (i.e. bigger differences in the number of dots between the patches); (top right) probability of choosing the left patch increased when the number of dots was higher in the patch at the left side ($\Delta Dots = Dots_{Left} - Dots_{Right}$); (bottom left) the probability of choosing an alternative depended on the gaze difference ($\Delta Gaze = g_{Left} - g_{Right}$); and (bottom right) the probability of choosing an item that was fixated longer than the other, corrected by the actual evidence $\Delta Dots$, depicted a residual effect of gaze on choice. Notice that in the free condition, the model predicted an effect of gaze on choice in a higher degree than in the fixed condition. Solid dots depict the mean of the data across participants in both conditions. Lighter dots present the mean value for each participant across bins. Solid grey lines show the average for model simulations. Data z-scored/binned for visualisation. (**D**) GLAM parameters fitted at participant level for free and fixed sampling conditions. Free sampling condition presented a higher gaze bias than fixed sampling, while no significant differences were found for the other parameters. $\gamma$: gaze bias; $\tau$: evidence scaling; $\nu$: drift term; $\sigma$: noise standard deviation. **: $p < 0.01$.

The online version of this article includes the following figure supplement(s) for figure 5:

**Figure supplement 1.** GLAM results with down-sampled gaze information in the free sampling condition.

**Figure supplement 2.** Individual out-of-sample GLAM predictions for behavioural measures in free and fixed sampling conditions.

**Figure supplement 3.** GLAM model comparison for free and fixed sampling conditions.

**Figure supplement 4.** GLAM parameters for free and fixed sampling conditions.

speed at which information is integrated, but it specifically modulates the likelihood of choosing the gazed-at option. Finally, to test that the identified effect did not depend on a less variable gaze time range we resampled the data from the free-sampling condition to match the gaze time range in the fixed-viewing condition, and fitted the GLAM again. We replicated our finding even when the gaze time range in the free-sampling condition was reduced to match that in the fixed-viewing condition (*Figure 5—figure supplement 1*). For more in-depth analyses including model comparisons please see *Figure 5—figure supplements 1–4*.

Finally, we explored the idea that a sampling bias could arise from the use of previous choice as evidence in its own right, in addition to previously acquired information. Formalizing this idea, we developed an economic model that makes behavioural predictions aligned with our findings (see Appendix 3). In this model, after the first choice, the prior that feeds into the subsequent active sampling phase is not the belief after the first sampling phase, but rather it is a convex combination of this with the belief that the individuals would hold if they only knew of the previous choice. Subsequent active sampling then varies as a strictly increasing function of this prior. As a result, a sampling bias in favour of the initially chosen option arises, and this bias varies as a function of confidence in the initial choice, as seen in our data. This economic model provides a formal description of the behaviour we observed. At the same time, it suggests that seeking for confirmatory evidence might not arise from a simple low-level heuristic, but is rooted in the way information is acquired and processed (for more details, see Appendix 3).

## Discussion

In this work, we have demonstrated that a form of confirmation bias exists in active information sampling, and not just in information weighting as previously shown. Using two novel experiments we showed that this effect is robust for simple perceptual choice. Critically we show that this sampling bias affects future choice. Furthermore, we demonstrated that this effect is only present in the free-sampling condition, showing that agency is essential for biased sampling to have an effect on subsequent choice.

Preference for confirmatory evidence has previously mostly been studied in the context of strongly held political, religious or lifestyle beliefs, and not in perceptual decision-making (*Bakshy et al., 2015*; *Hart et al., 2009*; *Lord et al., 1979*; *Nickerson, 1998*; *Stroud, 2007*; *Wason, 1960*, *Wason, 1968*). Our results, together with the recent work of others (*Talluri et al., 2018*; *Palminteri et al., 2017b*; *Lefebvre et al., 2020*), show that confirmation bias in information search is present even in decisions where the beliefs formed by participants are not meaningful to their daily lives. This suggests that confirmation bias might be a fundamental property of information sampling, existing irrespective of how important the belief is to the agent.

Recent findings suggest that biases in information sampling might arise from Pavlovian approach, a behavioural strategy that favours approaching choice alternatives associated with reward (*Hunt et al., 2016*; *Rutledge et al., 2015*). Furthermore, the number of hypotheses an individual can consider in parallel is likely to be limited (*Tweney et al., 1980*). As such, it may be advantageous to first attempt to rule out the dominant hypothesis before going on to sample from alternative options. In this vein, the sampling bias we see could be the solution to an exploit-explore dilemma in which the decision-maker must decide when to stop 'exploiting' a particular hypothesis (on which stimulus has the most dots) and instead 'explore' different ones.

A key novel feature of this task is that participants were able to freely sample information between choice phases, providing a direct read-out of confirmation bias in the active sampling decisions made by the participants. Previous accounts of confirmation bias in perceptual choice have instead focused on altered weighting of passively viewed information as a function of previous choice (*Bronfman et al., 2015*; *Rollwage et al., 2020*; *Talluri et al., 2018*). However, from these findings, it remained unclear to what extent this bias manifests in the processing of information compared to active sampling of information. Our findings show that active information sampling plays a key role in the amplification of beliefs from one decision to the next, and that changes in evidence weighting likely only account for part of observed confirmation bias effects.

We exclude some alternative explanations for our main findings. Firstly, it could have been possible for participants' biased sampling to have been driven by a need to reduce uncertainty determined by numerosity (*Scott et al., 2015*). It is unlikely that this was the case in this task, as neither total numerosity or numerosity of the chosen option significantly predicted sampling bias. Another possibility is that sampling bias is simply a measure of internal confidence, whereby sampling bias towards the initially chosen stimulus reflects the likelihood of it being chosen again. However, if this were the case a strong relationship between sampling bias and confidence in the second choice would be expected. We only see such a relationship in the first experiment, but not in the second experiment. This suggests sampling bias on this task cannot be fully attributed to an expression of endogenous confidence.

We show that confidence modulated the confirmation bias effect, such that choices made with higher confidence, led to increased sampling of the chosen option and an increased likelihood of choosing the same option again in the second choice phase. This shows that the strength with which a belief is held determines the size of the confirmation bias in active information sampling. Confidence has previously been shown to affect the integration of confirmatory evidence as reflected in MEG recordings of brain activity during evidence accumulation (*Rollwage et al., 2020*). Furthermore, recent work in economics and neuroscience have given theoretical and experimental proof of a relationship between overconfidence, and extreme political beliefs (*Ortoleva and Snowberg, 2015*; *Rollwage et al., 2018*). Our results suggest altered information sampling could be the missing link between these two phenomena. Specifically, given that we have shown that increased confidence leads to increased confirmation bias, it follows that overconfidence in a belief would lead to increased sampling of confirmatory evidence in line with that belief, which in turn would lead to even higher confidence.

We also show that it is not sufficient that agents freely make a choice for that choice to bias future decisions. When participants were presented with their chosen option for longer in a fixed sampling condition, this did not affect subsequent choice as it did in the free sampling condition, where a sampling bias in favour of the chosen option made changes of mind less likely. This suggests that integration of evidence into beliefs is dependent on whether the agent has actively chosen to sample this information. In other words, a sense of agency appears to impact the extent to which new information influences future choice (*Hassall et al., 2019*; *Chambon et al., 2020*; *Doll et al., 2011*; *Cockburn*

*et al., 2014*), and making a choice may not necessarily lead to confirmation bias if it is not followed by active decisions to sample evidence in line with that choice. Our results are in line with *Chambon et al., 2020*, in which confirmation bias in learning rates was only present when participants were able to freely choose between stimuli and not when these choices were fixed by the experimenter. In their task, though, choice and information gain were not separated by design, meaning information sampling was only possible from outcome feedback after a choice was made. Our results expand on these findings, by showing that choice commitment also affects subsequent decisions to sample, and that the resulting biased sampling influences future choices. This means that active information sampling is likely to play a central role in the propagation of confirmation bias across decisions, as seen in our descriptive economic model (see Appendix 3). It is less clear, however, to what extent the ability to freely sample and any resulting confirmation bias might be beneficial or detrimental to choice accuracy, as in our task there was no clear difference in accuracy between the free and fixed sampling conditions.

Additionally, the results from the GLAM model show that, in line with previous studies (*Krajbich et al., 2010*; *Krajbich et al., 2010*; *Sepulveda et al., 2020*; *Tavares et al., 2017*; *Thomas et al., 2019*), a specific boost in the accumulation of evidence of the visually attended items was found in the free sampling condition. Conversely, a disconnection between an item's sampling time and evidence accumulation was found in the fixed condition (i.e. the absence of gaze bias in GLAM implies that visual fixations did not affect evidence integration when the sampling was not controlled by the decision-maker). One explanation for this result is that attentional allocation itself is directed towards the options that the participants reckon more relevant for the task to be performed (*Sepulveda et al., 2020*). In our experiment, the goal of the task was theoretically identical for the first and second choice (i.e. to find the patch with more dots). However, in agreement with the ideas of our descriptive economic model, it could be that participants perceived the goal of the second choice to be slightly different from the goal of the first: in the second case they had to verify whether their initial choice was correct, as well as finding the patch with the most dots. This immediately bestowed the chosen option with higher relevance and then more attention (consequently boosting evidence accumulation). On the other hand, since in the fixed sampling condition participant's attention allocation is not necessarily associated with their goals, the difference in display time of the items is ignored or cannot be consistently integrated in their decision process. A complementary perspective has been given in recent work by *Jang et al., 2021* and *Callaway et al., 2021*. Their models characterize attention's role as lowering the variance of the accumulated evidence towards the attended option, which in turn updates the internal belief. Crucially, they have characterized attention as an internally generated factor that is allocated following optimal sampling policies, unlike other models that take attention as an exogenous factor (*Krajbich et al., 2010*; *Krajbich et al., 2010*; *Thomas et al., 2019*). Perhaps, in a similar way to the proposition above in relation to internal goals, it could be the case that the exogenous sampling pattern we imposed in our experiment is not aligned with the optimal evolution of the internal beliefs of participants, therefore, the 'offered' evidence is misaligned and therefore does not impact choice. Further modelling work testing Jang and Callaway proposals in regimes where attention is exogenously controlled can give insight on the relevance of attentional agency. An alternative hypothesis is that gaze allocation itself is used as additional evidence in favour of a choice alternative in a way similar to previous choice as in our descriptive model (see Appendix 3). This would mean that when an agent has previously attended to a choice alternative, this is used as evidence in favour of that option in and of itself. Therefore, when gaze allocation is not under the agent's control, as in the fixed-viewing condition, it is not used to inform choice.

Our findings imply that agency plays a clear role in evidence accumulation, and consequently in confirmation bias. It also suggests that common experimental designs in which information is provided by the experimenter and passively sampled by the participant might not be an ecological way to study decision-making. These tasks mask the potentially large effect of active information search on belief formation and choice. *Pennycook et al., 2021* recently showed that people were less likely to share false information online if they had been asked to rate the accuracy of a headline just previously. It may therefore be possible to reduce confirmation bias in information search in a similar way by priming participants to attend more to accuracy instead of confirmatory evidence. More complex behavioural tasks are required to improve our understanding of the different drivers of information sampling and how sampling in turn guides future choice (*Gottlieb and Oudeyer, 2018*).

To summarise our findings, we observed that participants sampled more information from chosen options in a perceptual choice paradigm and that this sampling bias predicted subsequent choice. Asymmetric sampling in favour of the chosen alternative was stronger the higher participants' confidence in this choice. Furthermore, the effect of information on subsequent choice was only seen in a version of the task where participants could sample freely, suggesting agency plays an important role in the propagation of strongly held beliefs over time. These findings suggest that confirmatory information processing might stem from a general information sampling strategy used to seek information to strengthen prior beliefs rather than from altered weighting during evidence accumulation only, and that active sampling is essential to this effect. Biased sampling may cause a continuous cycle of belief reinforcement that can be hard to break. Improving our understanding of this phenomenon can help us better explain the roots of extreme political, religious and scientific beliefs in our society.

## Materials and methods

### Participants

#### Experiment 1

Thirty participants took part in this study. We excluded two participants from the analysis, because they gave the highest possible confidence rating on more than 75% of trials. Participants received a £10 show-up fee in addition to a monetary reward between £0 and £6, which could be gained in the task. Participants had normal or corrected-to-normal vision, and no psychiatric or neurological disorders. We obtained written informed consent from all participants before the study. This experiment was approved by the University of Cambridge Psychology Research Ethics Committee.

#### Experiment 2

Twenty-three participants completed this experiment, of which two were excluded from the analysis, because they gave the highest possible confidence rating on more than 75% of trials. Another three participants were excluded because their confidence was a poor predictor of their accuracy in the task (lower than two standard deviations under the mean coefficient predicting accuracy in a logistic regression). Participants were reimbursed £30 for their time as well as an additional amount between £0 and £20 that could be gained in the task. Participants had normal or corrected-to-normal vision, and no psychiatric or neurological disorders. We obtained written informed consent from all participants before the study. This experiment was approved by the University of Cambridge Psychology Research Ethics Committee.

### Behavioural task

#### Experiment 1

The computer task used in this study consisted of 200 trials. On each trial, participants were asked to make binary choices between two dot patches. Stimuli were presented once for 700ms each in the first presentation phase, with the order of presentation assigned randomly. This was followed by the first choice phase, which was not time-restricted, in which participants could choose between the two stimuli using the arrow keys on the keyboard. After making a choice, participants rated their confidence on a continuous rating scale using the arrow keys, which was also not time-restricted. This was followed by the second phase in which participants could sample the stimuli freely for 4000ms by using the arrow keys to alternately view the stimuli. During this sampling phase, two empty circles were presented when participants did not press any of the buttons. To sample either of the stimuli, participants could press the left or right arrow key to make the dots appear in one of the circles. These dots would stay there until the participant pressed the other key, at which point the dots in the other circle would appear and those in the initially sampled circle would disappear. In other words, participants could never observe the two circles with dots at the same time. Participants could switch back and forth between the two stimuli as often they wanted within the allocated time. This was again followed by a choice phase and confidence rating, both of which were again not time-restricted. Participants were constantly reminded of their initial choice by a green tick next to the chosen patch. They received no feedback about the correctness of their choice. One of the patches always contained 50 dots, and the other a variable amount of dots. We calibrated the dot difference between the two patches such that accuracy level within each participant was maintained at 70%

throughout the task using a one-up two-down staircase procedure (*Fleming and Lau, 2014*). Trials were separated by a 1000ms inter-trial interval (ITI). The task was programmed using Python 2.7.10 and PsychoPy (*Peirce, 2007*).

## Experiment 2

A similar task was used in the second experiment. In this version, in the initial presentation phase each stimulus was presented for 500ms. Participants' responses were elicited through eye movements. To make a choice they looked at one of the patches and to rate their confidence they looked at a position inside a rating scale. The sampling phase between the two choices randomly varied in this task, and was either 3000, 5000, or 7000ms. This was not cued to the participants, so at the start of a sampling phase they did not know how much time they would be given to sample the patches. Sampling was completely gaze-contingent meaning that the dots in a circle were only visible when the participant fixated inside that circle and participants could only see one circle at a time. This was done by tracking when the participant's gaze was within one of two pre-defined square areas of interest (AI) centered on the two stimuli. Between each phase of the trial, participants had to fixate on a central fixation cross. Furthermore, we introduced a control condition in which participants were not free to sample the circles however they liked during the sampling phase. Instead, in one-third of trials the patches were shown for an equal amount of time each and in two-thirds of trials one patch was shown three times longer than the other (50% of trials the left was shown longer, 50% of trials the right was shown longer). Participants were constantly reminded of their initial choice by the circle surrounding the chosen patch changing color. Participants took part in two sessions, each consisting of 189 trials. In one session, they performed the main task and in the other the control condition of the task. The order of these two sessions was pseudo-random. This experiment was programmed using the SR Research Experiment Builder version 1.10.1630 (*SR Research Experiment Builder, 2017*).

### Incentive

The incentive structure used to ensure participants optimized their performance and indicated their true confidence was a linear scoring rule, where the number of points earned on each decision was as follows:

$$Points = 50 + correct * \left( \frac{confidence}{2} \right) \tag{1}$$

Here, the variable 'correct' is –1 for an incorrect choice and 1 for a correct choice. The variable 'confidence' is the confidence rating between 0 and 100. Each of the two choices on a trial was rewarded in this same way, and so should be considered equally important by participants. Participants received an additional £1 for every 3,780 points earned on the task. This scoring rule was made explicit to participants (see Appendix 4).

### Eye-tracking

In experiment 2, we recorded eye movements at a rate of 1000 Hz using an EyeLink 1000 Plus eye-tracker. Areas of Interest (AI) for the eye tracking analyses were pre-defined as two squares centered on the gaze-contingent circles in the experiment. The sides of the squares were the same as the diameter of the gaze-contingent circles. For each decision period, we derived the total dwell time in each AI from the eye-tracking data. The computer used in this experiment had a screen size of 68.58 × 59.77 cm and participants were seated 60 cm away from the screen.

### Analyses

We studied the effect of choice on the time spent on each of the two stimuli using paired sample t-tests on the mean sampling times spent on each stimulus from each participant. Trials with the shortest sampling phase length of 3000ms in experiment 2 were excluded from all analyses, because it became apparent that this time was too short for participants to be able to saccade to each circle more than once.

## Hierarchical models

Hierarchical regression models were conducted using the lme4 package in R (*Bates et al., 2015*; *Gelman and Hill, 2006*). All models allowed for random intercepts and slopes at the participant

level. We computed degrees of freedom and p-values with the Kenward-Roger approximation, using the package pbkrtest (**Halekoh and Højsgaard, 2014**). We predicted the sampling time difference between the two circles using a hierarchical linear regression model. To predict choice in the second choice phase, hierarchical logistic regressions were used predicting the log odds ratio of picking the left circle on a given trial. Confidence and sampling time were z-scored on the participant level. For detailed results and model comparisons, see Appendices 1, 2, 5.

## Attentional model - GLAM

The Gaze-weighted Linear Accumulator Model (**Thomas et al., 2019**; **Molter et al., 2019**) is part of the family of linear stochastic race models in which different alternatives (i; left or right) accumulate evidence (Ei) until a decision threshold is reached by one of them, determining the chosen alternative. The accumulator for an individual option was defined by the expression:

$$E_i\left(t\right) \ = \ E_i\left(t-1\right) \ + \ \nu R_i \ + \ \epsilon_t \ \ with \ \epsilon_t \sim N\left(0,\sigma\right) \ and \ E_i\left(t=0\right) = 0 \tag{2}$$

A drift term ($\nu$) controlled the speed of relative evidence ($R_i$) integration and noise was integrated with normal distribution (zero-centered and with standard deviation, σ). $R_i$ expressed the amount of evidence that was accumulated for item i at each time point t. This was calculated as follows. We denote by $g_i$, the relative gaze term, calculated as the proportion of time that participants observed item i:

$$g_i = \frac{DT_i}{DT_1+DT_2} \tag{3}$$

with DT as the dwelling time for item i during an individual trial. Let $r_i$ denote the evidence (in this study, evidence corresponded to number of dots presented for each option) for item i. We can then define the average absolute evidence for each item ($A_i$) during a trial:

$$A_i = g_i r_i + \left(1 - g_i\right)\gamma r_i \tag{4}$$

This formulation considers a multiplicative effect of the attentional component over the evidence, capturing different rates of integration when the participant was observing item i or not (unbiased and biased states, respectively). The parameter γ was the gaze bias parameter: it controlled the weight that the attentional component had in determining absolute evidence. When $\gamma = 1$, accumulation was the same in biased and unbiased states, that is gaze did not affect the accumulation of evidence. When $\gamma < 1$, $A_i$ was discounted for the biased condition, resulting in higher probability of choosing items that had been gazed at longer. When $\gamma < 0$, the model assumed a leak of evidence when the item was not fixated. Therefore, the relative evidence of item i, $R_i^*$, was given by:

$$R_i^* = A_i - \max_{j\neq i}A_j = A_i - A_j \ \rightarrow R_{right}^* = -R_{left}^* \tag{5}$$

Since our experiment considers a binary choice, while the original formulation of the model (**Thomas et al., 2019**) proposed for more than two alternatives, $R_i^*$ was reduced to subtract the average absolute evidence of the other item. Therefore, for the binary case, $R_i^*$ for one item was the additive inverse of the other. For example, if the left item had the lower evidence, we would have $R_{left}^* < 0$ and $R_{right}^* > 0$. The difference in signals captured by $R_i^*$ was scaled by a logistic transformation. The model assumed an adaptive representation of the relative decision signals, which is maximally sensitive to marginal differences in absolute decision signals:

$$R_i = \frac{1}{1+\exp\left(-\tau R_i^*\right)} \tag{6}$$

The temperature parameter $\tau$ of the logistic function controlled the sensitivity of the transformation. Larger values of $\tau$ indicate stronger sensitivity to small differences in absolute evidence ($A_i$). Given that $R_i$ represents an average of the relative evidence across the entire trial, the drift rate in $E_i$ can be assumed to be constant, which enables the use of an analytical solution for the passage of time density (for details see **Thomas et al., 2019**; **Molter et al., 2019**). Notice that unlike the aDDM (**Krajbich et al., 2010**), GLAM does not deal with the dynamics of attentional allocation process in

choice. In summary, the GLAM model considers four free parameters: $\nu$ (drift term), γ (gaze bias), $\tau$ (evidence scaling), and σ (normally distributed noise standard deviation).

The model fit with GLAM was implemented at a participant level in a Bayesian framework using PyMC3 (*Salvatier et al., 2016*). Uniform priors were used for all the parameters:

$v \sim \text{Uniform}(1^{-6}, 0.01)$
$\gamma \sim \text{Uniform}(-1, 1)$
$\sigma \sim \text{Uniform}(1^{-6}, 5)$
$\tau \sim \text{Uniform}(0, 5)$

We fitted the model for each individual participant and for free and fixed sampling conditions in experiment 2 separately. To model participants' behaviour, we used as input for GLAM the reaction times (RT) and choices obtained from phase 3, and relative gaze for left and right alternatives for each trial during sampling phase 2. For fixed sampling trials, the presentation times of the dot patches were used to calculate the relative gaze time. For both conditions, model fit was performed only on even-numbered trials using Markov-Chain-Monte-Carlo sampling, we used implementation for No-U-Turn-Sampler (NUTS), four chains were sampled, 1,000 tuning samples were used, and 2000 posterior samples were used to estimate the model parameters. The convergence was diagnosed using the Gelman-Rubin statistic ($|R - 1| < 0.05$) for the four parameters ($\nu$, γ, σ, and $\tau$). Considering all the individual models (18 participants), we found divergences in ~20% of the estimated parameters (~16% in free: ~ 25% in the fixed condition). We removed the participants that presented divergent parameters (7 participants) to check whether the results we found were driven by these data. The significantly higher gaze bias in free-viewing condition was maintained even after removing these participants (see *Figure 5—figure supplement 4*). Model comparison was performed using Watanabe-Akaike Information Criterion (WAIC) scores available in PyMC3, calculated for each individual participant fit. Note that in the fixed condition a model without gaze bias was more parsimonious than the model including the $\gamma$ parameter (*Figure 5—figure supplement 3*). The fact that the gaze model was not the most appropriate to capture the data in the fixed condition may explain why we observed more parameter divergences in that case.

To check how well the model replicates the behavioural effects observed in the data (*Palminteri et al., 2017b*), simulations for choice and RT were performed using participants' odd trials, each one repeated 50 times. For each trial, number of dots and relative gaze for left and right items were used together with the individually fitted GLAM parameters to simulate the trials. Random choice and RT (within a range of the minimum and maximum RT observed for each particular participant) were set for 5% of the simulations, replicating the contaminating process included in the model as described by *Thomas et al., 2019*.

## Acknowledgements

These studies were funded by the Wellcome Trust and Royal Society (Henry Dale Fellowship no. 102612/Z/13/Z to B.D.M.). P.S. was funded by the Chilean National Agency for Research and Development (ANID)/Scholarship Program/DOCTORADO BECAS CHILE/2017–72180193. The authors declare no conflict of interest.

## Additional information

### Funding

| Funder | Grant reference number | Author |
|---|---|---|
| Wellcome Trust | Henry Dale Fellowship | Benedetto De Martino |
| Royal Society | Henry Dale Fellowship | Benedetto De Martino |
| Chilean National Agency for Research and Development | Scholarship | Pradyumna Sepulveda |

| Funder | Grant reference number | Author |
|--------|------------------------|--------|

The funders had no role in study design, data collection and interpretation, or the decision to submit the work for publication.

## Author contributions

Paula Kaanders, Conceptualization, Data curation, Formal analysis, Investigation, Methodology, Project administration, Software, Validation, Visualization, Writing - original draft, Writing - review and editing; Pradyumna Sepulveda, Formal analysis, Methodology, Visualization, Writing - review and editing; Tomas Folke, Formal analysis, Software, Writing - review and editing; Pietro Ortoleva, Methodology, Writing - review and editing; Benedetto De Martino, Conceptualization, Funding acquisition, Methodology, Resources, Supervision, Writing - review and editing

## Author ORCIDs

Paula Kaanders http://orcid.org/0000-0002-5068-1946
Pradyumna Sepulveda http://orcid.org/0000-0003-0159-6777
Pietro Ortoleva http://orcid.org/0000-0002-5943-6621
Benedetto De Martino http://orcid.org/0000-0002-3555-2732

## Ethics

Human subjects: All participants signed a consent form and both studies were done following the approval given by the University of Cambridge, Cambridge Psychology Research Ethics Committee (PRE.2015.095).

## Decision letter and Author response

Decision letter https://doi.org/10.7554/eLife.71768.sa1
Author response https://doi.org/10.7554/eLife.71768.sa2

# Additional files

## Supplementary files

• Transparent reporting form

## Data availability

All data is available on the lab GitHub page (https://github.com/BDMLab).

The following dataset was generated:

| Author(s) | Year | Dataset title | Dataset URL | Database and Identifier |
|-----------|------|---------------|-------------|--------------------------|
| Kaanders P, Sepulveda P, Folke T, Ortoleva P, De Martino B | 2021 | Data from: Cherry-picking information: humans actively sample evidence to support prior beliefs | https://github.com/BDMLab/Kaanders_et_al_2021 | GitHub, GitHub |

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

# Appendix 1

## Hierarchical regression predicting sampling time difference

**Appendix 1—table 1.** Hierarchical Regression Model Predicting Sampling Time Difference, Experiment 1.

| Predictor | Coefficient | SE | t-value | DF | p-value |
|---|---|---|---|---|---|
| Intercept | −0.27 | 0.03 | −8.48 | 26.28 | < 0.0001 |
| Choice | 0.51 | 0.05 | 9.64 | 26.87 | < 0.0001 |
| Dot Difference | 0.05 | 0.004 | 13.02 | 26.80 | < 0.0001 |
| Confidence | −0.08 | 0.03 | −2.88 | 28.74 | 0.007 |
| Choice x Confidence | 0.19 | 0.04 | 5.26 | 26.59 | < 0.0001 |

This model was run using data from experiment 1. Confidence was z-scored per participant. It shows that choice is a significant predictor of sampling time difference between the chosen and unchosen circles, such that participants were likely to view the circle they chose for a longer time in the sampling phase. Furthermore, the interaction between choice and confidence was a significant predictor, signifying that the more confident participants were in their choice, the more biased their sampling was in favour of that choice. Dot difference also significantly predicts sampling time: participants sampled the circle containing the most dots for longer. Note that this variable shares some variance with the choice variable.

**Appendix 1—table 2.** Hierarchical Regression Model Predicting Sampling Time Difference, Experiment 2.

| Predictor | Coefficient | SE | t-value | DF | p-value |
|---|---|---|---|---|---|
| Intercept | −0.15 | 0.06 | −2.63 | 16.94 | 0.017 |
| Choice | 0.30 | 0.10 | 2.90 | 16.75 | 0.01 |
| Dot Difference | 0.04 | 0.005 | 9.21 | 68.14 | < 0.0001 |
| Confidence | −0.09 | 0.04 | −2.43 | 18.31 | 0.03 |
| Choice x Confidence | 0.21 | 0.05 | 4.29 | 32.00 | 0.0005 |

We also ran this model using data from experiment 2, excluding control condition trials in which sampling time difference was fixed. Confidence was z-scored per participant. Again, choice, the interaction term between choice and confidence and dot difference were all significant predictors of the difference in sampling time between the two circles.

# Appendix 2

## Hierarchical logistic regression predicting change of mind

**Appendix 2—table 1.** Hierarchical Logistic Regression Model Predicting Change of Mind, Experiment 1.

| Predictor | Coefficient | SE | z-value | p-value |
|---|---|---|---|---|
| Intercept | −1.18 | 0.21 | −5.65 | < 0.0001 |
| Sampling Bias | −1.24 | 0.08 | −11.0 | < 0.0001 |
| Dot Difference | −0.24 | 0.11 | −3.06 | 0.002 |
| Confidence | −0.58 | 0.06 | −10.12 | < 0.0001 |

This model was run using data from experiment 1. Using a hierarchical logistic regression we predicted the log odds ratio that the participant changed their mind in the second decision phase on a given trial. Sampling bias and confidence were z-scored per participant. Convergence issues were addressed by square-root-transforming dot difference. Sampling bias is defined as the difference between the amount of time the initially chosen vs unchosen circles were sampled. This variable negatively predicts change of mind, meaning the more the chosen circle was sampled, the less likely participants were to change their mind. Absolute dot difference, which is a measure of trial difficulty, negatively predicts change of mind, such that participants were less likely to change their mind when the trial was easier. Finally, confidence was a negative predictor of change of mind; the higher confidence in the initial choice, the less likely participants were to change their mind.

**Appendix 2—table 2.** Hierarchical Logistic Regression Model Predicting Change of Mind, Experiment 2.

| Predictor | Coefficient | SE | z-value | p-value |
|---|---|---|---|---|
| Intercept | −1.05 | 0.28 | −3.76 | 0.0002 |
| Sampling Bias | −1.39 | 0.19 | −7.20 | < 0.0001 |
| Dot Difference | −0.25 | 0.10 | −2.66 | 0.008 |
| Confidence | −0.57 | 0.07 | −8.73 | < 0.0001 |
| Fixed Sampling | −0.09 | 0.14 | −0.61 | 0.54 |
| Sampling Bias x Fixed Sampling | 1.30 | 0.19 | 6.77 | < 0.0001 |

A similar model to that presented was run on the data from experiment 2. Because this experiment included a control condition in which sampling was fixed, we included a dummy variable in this model coding whether the trial was in the control condition or not. Sampling bias, dot difference, and confidence were all significant negative predictors of change of mind. However, an interaction term between 'Fixed Sampling' (the control condition) and sampling bias was significantly positive, and approximately the same size as the main effect of sampling bias on change of mind. This means that in control condition trials, the effect of sampling bias on change of mind disappears.

# Appendix 3

## A Descriptive Model of Confirmatory Information Processing

We have designed the following descriptive economic decision-making model that can be used to capture the findings described above. There is a set of states of the world that denote the number of dots in each part of the screen, denoted $\left(\omega^L, \omega^R\right) \in R_+^2$. In what follows, for any probability measure μ over $R_+^2$, denote by $\mu\left(\omega^L > \omega^R\right)$ the probability that μ assigns to the event that $\omega^L$ is above $\omega^R$; and by $BU\left(\mu|A\right)$ the Bayesian update of μ using information $A$. Subjects start with a symmetric prior $\mu_0$.

**First Stage**: Subject go through a sampling phase in which they gather information about the number of dots in each screen. They get two noisy signals about each component of the state, $x_1^L$ and $x_1^R$, for which for simplicity we assume normally distributed noise:

$$x_1^L = \omega^L + \epsilon_1^L$$

$$x_1^R = \omega^R + \epsilon_1^R$$

where $\epsilon_1^L \sim N\left(0, \sigma_1^L\right)$, $\epsilon_1^R \sim N\left(0, \sigma_1^R\right)$. For generality, we may allow the variances to be different, in case individuals follow an adaptive search procedure that leads to asymmetric information acquisition. For simplicity, we assume here that they are the same, $\sigma_1^L = \sigma_1^R = \sigma^2$.

At the end of the information gathering stage, subjects form a posterior about the state of the world, $\widehat{\mu_1} = BU\left(\mu_0|x_1^R, x_1^L\right)$. They are then asked to choose an action $a_1 \in \{L, R\}$ and a confidence level $c$. Following standard arguments, they choose $a_1 = L$ if $\widehat{\mu_1}\left(\omega^L > \omega^R\right) > 0.5$, $a_1 = R$ if $\widehat{\mu_1}\left(\omega^L > \omega^R\right) < 0.5$, and randomize with equal probability between $L$ and $R$ if $\widehat{\mu_1}\left(\omega^L > \omega^R\right) = 0.5$ (a probability 0 event).

They report a confidence

$$c = 2 * \left|\widehat{\mu_1}\left(\omega^L > \omega^R\right) - \tfrac{1}{2}\right|.$$

Note that $c = 1$ if $\widehat{\mu_1}\left(\omega^L > \omega^R\right) \in \{1, 0\}$, $c = 0$ if $\widehat{\mu_1}\left(\omega^L > \omega^R\right) = 0.5$.

**Second Stage**: The key departure from the standard normative model of decision-making is that the beliefs that individuals have at the beginning of stage 2 are not the posteriors they obtained at the end of stage 1, but are also influenced by their choice. Denote $\widehat{\mu_1}^D$ the belief over the state of the world that the individual would obtain if she only observed her choice – given the decision rule above – but did not take into account the signals $x_1^L$ and $x_1^R$, that is, $\widehat{\mu_1}^D = BU\left(\mu_0|a_1\right)$. We posit that the belief used by the individual at the beginning of the second stage, $\mu_2$, is a convex combination of this belief and the posterior at the end of period 1, that is,

$$\mu_2 = \theta\widehat{\mu_1}^D + \left(1 - \theta\right)\widehat{\mu_1}$$

where $\theta \in \left[0, 1\right]$ indicates the degree of distortion: when $\theta = 0$ beliefs are not distorted; when $\theta = 1$, choices in the second stage only use the more limited information contained in the choice of stage 1, and not the full information coming from observed signals. The use of the linear structure is not relevant but simplifies the analysis.

In the second stage, subjects have to decide the sampling strategy. We posit that the fraction of time spent on Left is determined (with noise) as a strictly increasing function of the belief $\mu_2$: subjects spend more time on average on the area where they believe are more dots. In *Sepulveda et al., 2020* a model is described that generates a related tendency. In particular, we posit that the fraction of time spent on the left circle, $s$, is determined by a standard logistic function with parameter $k$

$$s = \frac{1}{1+e^{-k\left(\mu_2\left(\omega^L>\omega^R\right)-0.5\right)}}.$$

Through this sampling, individuals receive signals

$$x_2^L = \omega^L + \epsilon_2^L$$

$$x_2^R = \omega^R + \epsilon_2^R$$

where $\epsilon_1^L \sim N\left(0, \frac{\sigma^2}{s}\right), \epsilon_2^R \sim N\left(0, \frac{\sigma^2}{1-s}\right)$, so that better information is obtained about each dimension the more time spent contemplating it. Using this, subjects form a posterior $\widehat{\mu_2} = BU\left(\mu_2|x_2^L, x_2^R\right)$.

Finally, subjects choose an action $a_2 \in \{L, R\}$ and a confidence level $c$. Following standard arguments, they choose $a_1 = L$ if $\widehat{\mu_2}\left(\omega^L > \omega^R\right) > 0.5$, $a_1 = R$ if $\widehat{\mu_2}\left(\omega^L > \omega^R\right) < 0.5$, and randomize with equal probability between $L$ and $R$ if $\widehat{\mu_2}\left(\omega^L > \omega^R\right) = 0.5$ (again, a 0 probability event).

This model is reminiscent of the Second-Order Model of *Fleming and Daw, 2017*, but with the difference that we are not introducing a choice-affecting-beliefs component at the time of reporting confidence, but only at the beginning of the second stage.

This model implies that, on average, subjects will sample more from the patch they had previously chosen; and that, as long as $\theta \in (0, 1)$, this effect is stronger the higher their confidence in their choice—which is what we see in the data.

When $\theta > 0$, the previous choice affects the sampling strategy even controlling for confidence—which is also what we find. Whenever $\widehat{\mu_1}^D\left(\omega^L > \omega^R\right) > \widehat{\mu_1}\left(\omega^L > \omega^R\right) > 0.5$, the presence of $\theta > 0$ will lead to $\mu_2\left(\omega^L > \omega^R\right) > \widehat{\mu_1}\left(\omega^L > \omega^R\right) > 0.5$. This, in turn, will distort her later sampling strategy as well as her future beliefs: the individual will require even stronger information to change her mind. The opposite happens when $\widehat{\mu_1}\left(\omega^L > \omega^R\right) > \widehat{\mu_1}^D\left(\omega^L > \omega^R\right) > 0.5$: in this case $\widehat{\mu_1}\left(\omega^L > \omega^R\right) > \mu_2\left(\omega^L > \omega^R\right) > 0.5$, which implies that the individual acts as if she were less confident of her choice. Previous choice will affect the sampling strategy, but now less so.

# Appendix 4

## Scoring rule instructions for participants

For each of the two sessions you get £15 for participating and up to £10 extra based on your performance, meaning that your total payment for the experiment would range between £30 and £50.

Your performance will be measured in points, which are calculated based on your accuracy and confidence. The graph below illustrates how the points are calculated: the green line indicates the number of points you get as a function of your confidence rating if your decision is correct, and the red line indicates the number of points you get when your decision is incorrect.

This rule can be expressed mathematically as: points = 50 + correct × (confidence÷2)

The variable 'correct' is –1 for an incorrect choice and 1 for a correct choice. The variable 'confidence' is your confidence rating ranging from 0 to 100.

For example, say your first decision is correct (correct = 1) and you feel pretty confident about your choice (80 on the scale). You would then receive 50 + 1 × (80÷2) = 90 points. In contrast, say you picked the wrong option and were less confident (20) in your choice. You would then receive 50 + –1 × (20÷2) = 40 points.

Therefore, the more accurate you are and the better your confidence matches the correctness of your decision, the more money you earn. 3,780 points correspond to £1, and points will be converted to the nearest 10 pence. You will be paid by cheque within 4 weeks of completing the second experimental session.

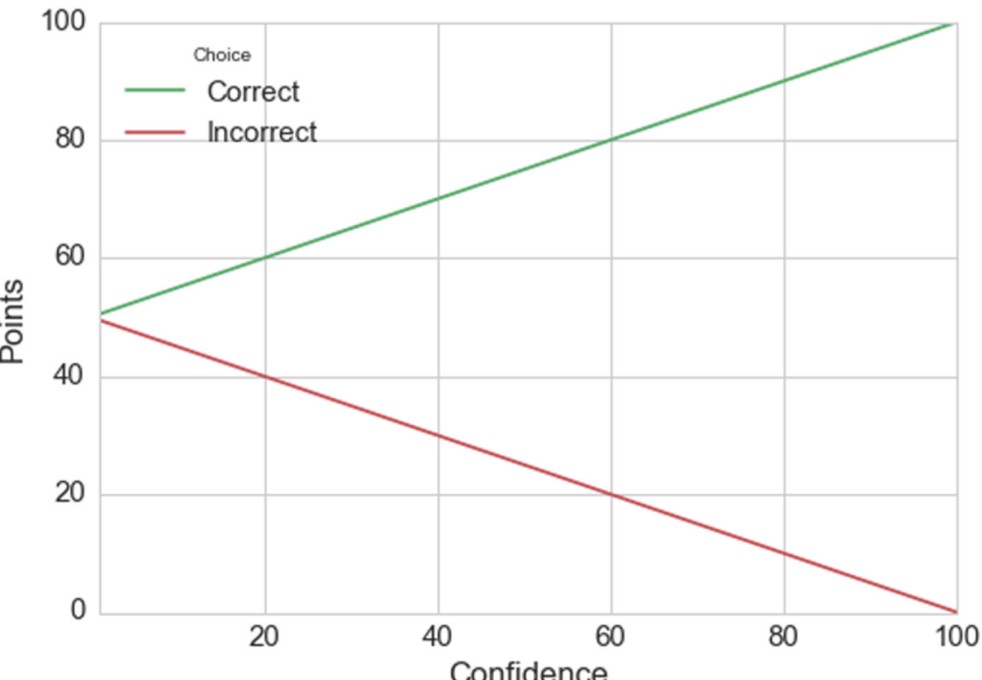

**Appendix 4—figure 1.** Linear scoring rule.

## Appendix 5

### Model Comparisons
#### Sampling Time Models
To investigate the effects of choice, dot difference, confidence, and response time on sampling we compared 6 hierarchical regression models. These models are presented below. The BIC scores for each model in each experiment are plotted in *Appendix 4—figure 1*. The best fitting models according to BIC-scores were Model 5 in experiment 1 and Model 3 in experiment 2. We chose to present Model 5 for both experiments as we were interested in the contribution of confidence to biased sampling.

**Appendix 5—table 1.** Sampling Time Models.

| Models | Formula |
| --- | --- |
| 1 | Sampling Time Difference $\sim \mathcal{N}(\beta_0 + \beta_1[\text{Choice}] + \varepsilon)$ |
| 2 | Sampling Time Difference $\sim \mathcal{N}(\beta_0 + \beta_1[\text{Dot Difference}] + \varepsilon)$ |
| 3 | Sampling Time Difference $\sim \mathcal{N}(\beta_0 + \beta_1[\text{Choice}] + \beta_2[\text{Dot Difference}] + \varepsilon)$ |
| 4 | Sampling Time Difference $\sim \mathcal{N}(\beta_0 + \beta_1[\text{Choice}] + \beta_2[\text{Confidence}] + \beta_3[\text{Choice} * \text{Confidence}] + \varepsilon)$ |
| 5 | Sampling Time Difference $\sim \mathcal{N}(\beta_0 + \beta_1[\text{Choice}] + \beta_2[\text{Dot Difference}] + \beta_3[\text{Confidence}] + \beta_4[\text{Choice} * \text{Confidence}] + \varepsilon)$ |
| 6 | Sampling Time Difference $\sim \mathcal{N}(\beta_0 + \beta_1[\text{Choice}] + \beta_2[\text{Dot Difference}] + \beta_3[\text{Confidence}] + \beta_4[\text{Reaction Time}] + \beta_5[\text{Choice} * \text{Confidence}] + \beta_6[\text{Choice} * \text{Reaction Time}] + \varepsilon)$ |

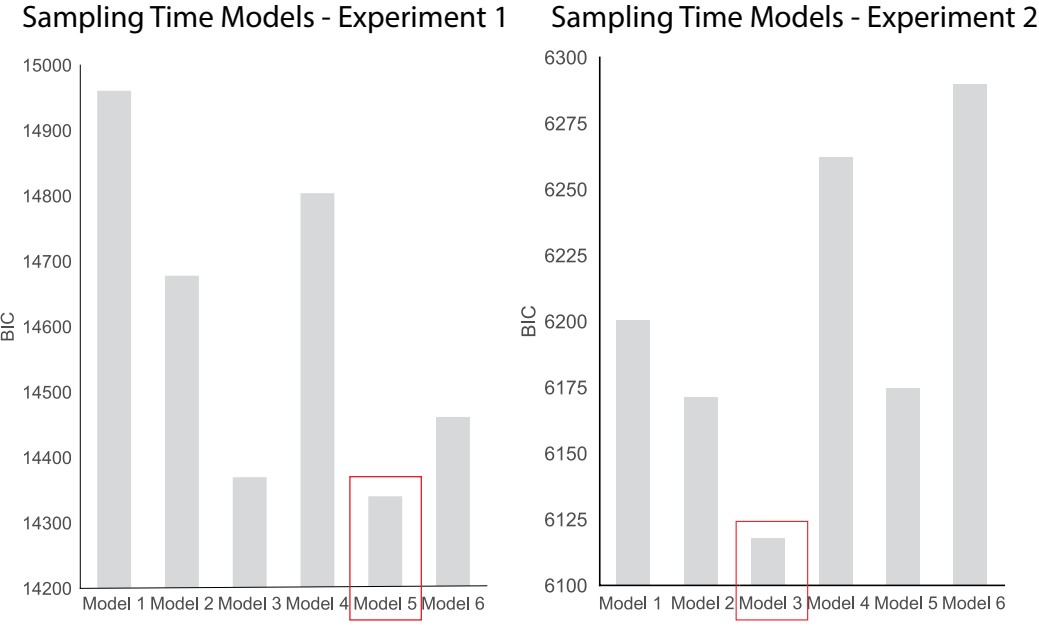

**Appendix 5—figure 1.** BIC comparison of the sampling time models for experiments 1 and 2.

Model 5 fit the data from experiment 1 the best (BIC = 14340.8), whereas Model 3 was the best fit for the data in experiment 2 (BIC = 6117.8).

#### Change of Mind Models
To investigate the effects of dot difference, sampling, confidence, and response time on change of mind we compared 5 hierarchical regression models. These models are presented below. The BIC scores for each model in each experiment are plotted in *Appendix 5—figure 2*. Model 5 includes

a dummy variable coding whether or not the trial was in the control condition or not (in which sampling was fixed). As such, this model is only applicable to experiment 2. The best fitting models according to BIC-scores were Model 3 in experiment 1 and Model 5 in experiment 2.

## Change of Mind Models

| Models | Formula |
|---|---|
| 1 | Change of Mind ~ $logit^{-1}(\beta_0 + \beta_1[Dot\ Difference] + \epsilon)$ |
| 2 | Change of Mind ~ $logit^{-1}(\beta_0 + \beta_1[Dot\ Difference] + \beta_2[Sampling\ Bias] + \epsilon)$ |
| 3 | Change of Mind ~ $logit^{-1}(\beta_0 + \beta_1[Dot\ Difference] + \beta_2[Sampling\ Bias] + \beta_3[Confidence] + \epsilon)$ |
| 4 | Change of Mind ~ $logit^{-1}(\beta_0 + \beta_1[Dot\ Difference] + \beta_2[Sampling\ Bias] + \beta_3[Confidence] + \beta_4[Reaction\ Time] + \epsilon)$ |
| 5 | Change of Mind ~ $logit^{-1}(\beta_0 + \beta_1[Dot\ Difference] + \beta_2[Sampling\ Bias] + \beta_3[Confidence] + \beta_4[Fixed\ Sampling] + \beta_5[Sampling\ Bias * Fixed\ Sampling] + \epsilon)$ |

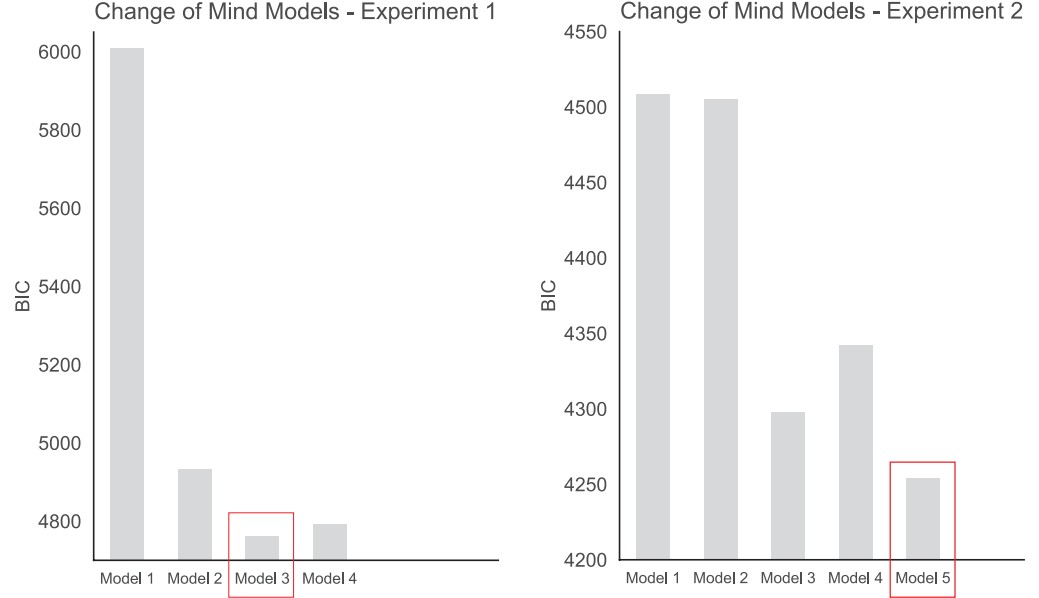

**Appendix 5—figure 2.** BIC comparison of the change of mind models for experiments 1 and 2. Model 3 fit the data from experiment 1 the best (BIC = 4760.8), whereas Model 5 was the best fit for the data in experiment 2 (BIC = 4251.7).

