## [Editor Report]

Kaanders et al. investigate how the sampling of visual information by human subjects is biased toward their previous choice. The novel experiments and rigorous analyses largely support the presence of a "confirmation bias" that arises specifically when information sampling is under subjects' control. These findings should be of interest to a broad community ranging from decision-making to metacognition research.

---

## [Decision Letter]

**Decision letter after peer review:**

Thank you for submitting your article "Cherry-picking information: humans actively sample evidence to support prior beliefs" for consideration by *eLife*. Your article has been reviewed by 3 peer reviewers, one of whom is a member of our Board of Reviewing Editors, and the evaluation has been overseen by Michael Frank as the Senior Editor. The following individual involved in review of your submission has agreed to reveal their identity: Konstantinos Tsetsos (Reviewer #2).

All reviewers provided a positive evaluation of the manuscript, and found that your work should be of interest to a broad research community ranging from decision-making to metacognition. The finding that the confirmation bias you observe in your study depends critically on active information sampling is interesting and novel. This finding is supported by rigorous analyses of the behavioral data, replicated across two distinct datasets. Nevertheless, we have identified different revisions that should be made to the current version of the manuscript to clarify aspects of the task, report additional control analyses, and rule out alternative explanations of the observed behavior. Essential revisions, for which you should provide a point-by-point response along with your revised manuscript, are provided first. Other points, which you can use to further clarify aspects of the study but do not need a point-by-point response, are provided afterwards.

Essential revisions:

1. Modeling. Influential theories of information sampling posit that sampling aims at reducing uncertainty – in your task, uncertainty regarding which option has more dots. This uncertainty-minimizing sampling (sometimes coined as 'directed exploration' in the literature) strategy is actually the optimal strategy in your task (i.e., the strategy that maximizes the accuracy of the second decision). By contrast, your economic model of information sampling hypothesizes that human agents seek to sample from the option associated with the largest perceived number of dots – a suboptimal strategy for maximizing the accuracy of the second decision. If subjects are sampling from the option which they perceive to have more dots (as in your model), then the sampling bias toward the previously chosen option reflects the confirmation bias that you describe. However, if subjects aim at reducing uncertainty through sampling (the optimal strategy), then the same sampling bias does not reflect a confirmation bias. It could instead reflect the fact that participants are actually minimizing uncertainty without any active choice bias. Indeed, research in numerical cognition has shown that internal noise scales positively with numerosity. Therefore, it could be the case that participants happen to sample more from the option associated with the largest perceived number of dots (the previously chosen one) because it is associated with higher uncertainty (larger internal noise). The authors should rule out this alternative explanation of the observed sampling bias in the revised manuscript. We have identified different ways to achieve this. Using new data (not required), a new condition in which subjects are instructed to select the alternative with fewer dots would help: indeed, a sampling bias toward the previously chosen option in this new condition would rule out the alternative explanation. Performing a direct comparison between the two sampling strategies (accounting for the positive scaling of internal noise with numerosity) and showing that human sampling is better explained by a suboptimal sampling in direction of the option perceived to be associated with more dots would also work. The authors could also possibly rule out this alternative explanation by relying on the confidence data during the second decision. In any case, the authors should consider the possibility that subjects may be sampling from the option that minimizes uncertainty about which option has the largest number of dots. In doing so, the modeling of the observed behavior should be expanded in the revised manuscript by describing it and discussing it in the main text, rather than only in the Supplemental Material section.

2. Task. Some important details concerning the task are missing. In the first phase, it is not clear how the dot patches were delivered to the participants. Figure 1A shows two stimulus presentation phases but the details on the duration of each phase, or whether there was a gap between phases, are missing. In the second phase, the authors should explain more clearly in the task how the sampling phase works, i.e. what the temporal dynamics of responding are. Do subjects press the left key and see the left stimulus until they press right, or does each key press display the stimulus for a period of time? Does the sampling phase start showing one side chosen randomly by the computer, or do subjects need to press a key to start seeing one stimulus? More details about the gaze-contingency and monitoring used in Experiment 2 would help the reader understand how this is done, as would reporting an analysis of the observed looking times for the chosen vs unchosen stimuli in the free vs. fixed sampling conditions (even if only in the Supplementary Material).

3. Task. Some aspects of the task deserve further description and discussion. In the first phase, the sequential presentation of the two choice-stimuli invites questions about temporal order biases, both in terms of the initial choice and in terms of the subsequent sampling (how do decision weights change across time). Do people show a choice and a sampling bias toward the first (or last) presented alternative? The confirmation bias effect could occur spuriously if, say, people tend to choose the early alternative and tend to also sample from the early presented alternative due to having forgotten that information (e.g., because of a temporal leak in evidence accumulation). In the second phase, highlighting the chosen alternative with a green arrow could bias the subsequent post-choice sampling towards the chosen alternative (and thus explain part of the observed effect). The authors should either mention this possibility, or state why it is unlikely to drive the observed effect.

4. Contrast between active and passive sampling conditions in Experiment 2. In the passive (fixed) condition, the first choice seems to be less meaningful given that the subjects know that they will subsequently observe more evidence. It would be important to show the psychometric data in Figure 3B separately for active (free) and passive (fixed) sampling and examine if there are any differences between the two conditions (e.g. choice 1 in the fixed sampling condition might be characterised by lower sensitivity/accuracy). Regarding changes-of-mind, is there any evidence that the fixed sampling condition helped counteract potential errors in the first choice, e.g., in cases in which the confirmation-biased free sampling might have led to further committing to an erroneous choice (as suggested in the abstract and the description of the confirmation bias), but the fixed sampling increased the likelihood of changing their mind? Along the same lines, it would be very helpful to know the fraction of changes-of-mind in the two conditions, so that readers can appreciate whether the fixed sampling condition leads to an increase in changes-of-mind. Last, the fact that the effect of sampling bias on changes-of-mind is not there in the fixed sampling condition may not necessarily mean that it is agency itself which drives the effect. Under the authors' hypothesis that subjects aim at sampling from the option with the largest perceived number of dots, the sampling bias measured in the free condition may reflect an endogenous/internal confidence signal regarding the first choice – an internal signal that predicts how likely it is that subjects will confirm this first choice in the second choice. In the fixed condition, this endogenous/internal confidence signal cannot be measured because sampling is controlled by the experimenter, and can only appear as background noise when modeling the second choice. In this view, agency is not driving the effect in a cognitive sense, but only allowing for the effect of an endogenous/internal confidence signal to be detected. This alternative account should either be ruled out by additional analyses or discussed in the main text.

5. Analysis of confidence ratings for the second choice. Since confidence ratings were also obtained at the second choice, it seems it would be interesting to analyse those to assess whether these are similarly influenced specifically by actively biased information sampling, e.g. allowing people to feel even more confident in the second choice, but also how that interacts with potential changes of mind. The analysis of confidence ratings for the second choice may also help rule out the possibility that the sampling bias observed in the free sampling condition is driven not by a biased perception of the number of dots toward the previously chosen option, but by an increased internal noise (and larger uncertainty) for higher numerosities (see point 1).

6. The illustration of the relation between confidence and sampling bias for the chosen option (Figure 2A and Figure 4A) is confusing. Currently, the curve in Figure 2A seems quite non-monotonic. It is therefore unclear how the conclusion in the title of this figure panel is derived. More precisely, what is confusing is that the correlation stated is an increase in confidence being linked to increased sampling bias, but the data plotted on Figure 2A seem to depict an opposing pattern. This may be because the extreme confidence values show the opposite pattern to intermediate levels, suggesting there isn't a simple linear relation. In fact, the main effect of confidence is negative (in both experiments), which is currently not discussed. What is also unclear is whether this plot is already depicting the choice x confidence interaction? It would be very useful to clarify this illustration, and describe explicitly in the main text whether the model is predicting the sampling time difference between the chosen and unchosen options, or something else (e.g., between the left and right options).

7. Feedback. It is not clear why feedback was not provided on a trial-by-trial basis. More generally, how were participants incentivised in the reported experiment? Did they have reasons to assign equal importance to the first and second choices? More details should be provided regarding these important features of the experiment – given the economic model proposed by the authors makes the assumption that subjects aim at sampling from the option associated with the largest number of dots (a suboptimal strategy) rather than reducing the uncertainty about which option has the largest number of dots (the optimal strategy in the context of the current task).

*Reviewer #1 (Recommendations for the authors):*

– The current illustration of the relation between confidence and sampling bias in Figure 2A (and Figure 4A) is misleading. This is probably due to the binning strategy (in bins of fixed size on the x-axis rather than fixed size in number of data points), but the relation described by the authors in the main text does not correspond to the apparent relation shown on Figure 2A, due to the extreme bins (likely to be strongly underpopulated). The authors should use a different binning strategy so that it is less affected by outliers.

– In Experiment 2, the fact that the effect of sampling bias on changes-of-mind is not there in the passive (fixed) condition may not necessarily mean that it is the active sampling process itself which drives (causes) the effect. Under the authors' hypothesis that subjects aim at sampling from the option with the largest perceived number of dots (in the free condition), the sampling bias measured in the free condition may reflect an endogenous/internal confidence signal regarding the first choice, an internal signal that then predicts how likely it is that subjects will confirm this first choice in the second choice. In the fixed condition, this endogenous/internal confidence signal cannot be measured because sampling is controlled by the experimenter, and will then only appear as background noise in statistical models of the second choice. In this view, agency (active sampling) is not driving the effect reported by the authors in a cognitive sense, but only allowing for the effect of an endogenous/internal confidence signal to be measured. This alternative account should be either ruled out by additional control analyses or discussed in the revised manuscript.

*Reviewer #2 (Recommendations for the authors):*

- Please provide more details about the experimental paradigm.

- The curve in Figure 2A seems quite non-monotonic. It is not clear how the conclusion in the title of this figure is derived.

- Related to the public comments above, there could be a confound between numerosity and post-choice sampling. It could be instructive to plot the proportion of total time (Figure 1C) separately for correct and incorrect initial decisions.

- A recent paper by Jang, Sharma and Drugowitsch (*eLife*, 2021) could be relevant for discussion.

- Similar to the "predicting sampling time difference" (Figure 2B) the authors could perform a similar analysis to predict the total post-choice sampling time (assuming that this is not always fixed to 4000 ms).

*Reviewer #3 (Recommendations for the authors):*

In addition to considering the points raised in the public review, the authors should consider the following revisions:

- Exp 1 – should explain more clearly in the task how the sampling phase works, i.e. what's the temporal dynamics of responding? Do they press the left key and see the left stimulus until they press right, or each key press display the stimulus for x time? Does the sampling phase start showing one side chosen randomly by the computer, or do they need to press a key to start seeing one stimulus?

- Exp 1 , p.5 – Figure 2 A – seems confusing that the correlation stated is an increase in confidence being linked to increased sampling bias, but the data plotted seem to depict an opposing pattern (pattern looks more consistent for Exp 2, in Figure 4A). This may be because the extreme confidence values show the opposite pattern to intermediate levels, suggesting there isn't a simple linear relation. In fact, the main effect of confidence is negative (in both experiments), which isn't discussed. But it's also unclear to me if that plot is already depicting the choice x confidence interaction? I.e. please clarify whether the model is predicting the sampling time difference between chosen – unchosen (DV in the plot 2A), or e.g. left-right?

- Hierarchical regression models – for clarify and reproducibility, please describe in the text how the varying (random) effects were specified (i.e. random intercepts and slopes?), and include the calculated degrees of freedom in the supplementary tables.

- P. 16, l.461-462: "We do not see an effect of the first choice on subsequent choices when the intermediate sampling phase is fixed by the experimenter" – it's unclear to me what is meant by this, or what evidence is taken to demonstrate that the first choice did not influence the second? People are generally unlikely to change their minds. Is there any evidence that the fixed information sampling conditions helped counteract potential errors in the first choice, e.g. in cases in which the confirmation-biased free sampling might have led to further committing to an erroneous choice, but the fixed sampling increased the likelihood of changing their mind?

- P. 15, l. 424-426 "might arise from Pavlovian approach, a behavioural strategy that favours approaching choice alternatives associated with reward" – seems unclear how this is relevant here since no rewards are involved. If the authors are hinting at the proposal that freely-made choices are inherently rewarding (e.g. Leotti and Delgado 2011), then it would be helpful to make that link explicit, or further explain the intended idea.

[Editors' note: further revisions were suggested prior to acceptance, as described below.]

Thank you for resubmitting your work entitled "Humans actively sample evidence to support prior beliefs" for further consideration by *eLife*. Your revised article has been evaluated by Michael Frank (Senior Editor) and Valentin Wyart (Reviewing Editor).

You have provided convincing and detailed responses to most comments. The revised manuscript is stronger and provides a more adequate discussion of the findings.

Nevertheless, there are two points that we think should be addressed before considering the manuscript suitable for publication at *eLife*.

1. Additional information about the surprising sampling patterns in the fixed condition of Experiment 2. You describe in your response to essential revisions that participants are actually looking at the stimulus presented for a shorter period of time much longer than the stimulus presented for a longer period of time. The graph provided shows that participants look at one stimulus much longer than its presentation time, indicating that participants look inside the blank placeholder while the other stimulus is being presented. This is not central to the paper's findings, but the fixed condition being a control for the free condition, it is important to understand why it may be the case – given the strength and apparent irrationality of the effect. We would like to know how this effect interacts not with chosen/unchosen, but with whether the stimulus presented for shorter was presented first or second. The current explanation for the effect offered in the response letter is currently unclear.

2. You describe the incentive structure as if it is fully known by the participants. However, it is unclear whether it is indeed the case during the instructions phase. We would like you to clarify whether or not the incentive structure of the task was made explicit to the participants, or whether you only use this incentive structure to derive theory-based interpretations of the observed effects.

---

## [Author Response]

Essential revisions:1. Modeling. Influential theories of information sampling posit that sampling aims at reducing uncertainty – in your task, uncertainty regarding which option has more dots. This uncertainty-minimizing sampling (sometimes coined as 'directed exploration' in the literature) strategy is actually the optimal strategy in your task (i.e., the strategy that maximizes the accuracy of the second decision). By contrast, your economic model of information sampling hypothesizes that human agents seek to sample from the option associated with the largest perceived number of dots – a suboptimal strategy for maximizing the accuracy of the second decision. If subjects are sampling from the option which they perceive to have more dots (as in your model), then the sampling bias toward the previously chosen option reflects the confirmation bias that you describe. However, if subjects aim at reducing uncertainty through sampling (the optimal strategy), then the same sampling bias does not reflect a confirmation bias. It could instead reflect the fact that participants are actually minimizing uncertainty without any active choice bias. Indeed, research in numerical cognition has shown that internal noise scales positively with numerosity. Therefore, it could be the case that participants happen to sample more from the option associated with the largest perceived number of dots (the previously chosen one) because it is associated with higher uncertainty (larger internal noise). The authors should rule out this alternative explanation of the observed sampling bias in the revised manuscript. We have identified different ways to achieve this. Using new data (not required), a new condition in which subjects are instructed to select the alternative with fewer dots would help: indeed, a sampling bias toward the previously chosen option in this new condition would rule out the alternative explanation. Performing a direct comparison between the two sampling strategies (accounting for the positive scaling of internal noise with numerosity) and showing that human sampling is better explained by a suboptimal sampling in direction of the option perceived to be associated with more dots would also work. The authors could also possibly rule out this alternative explanation by relying on the confidence data during the second decision. In any case, the authors should consider the possibility that subjects may be sampling from the option that minimizes uncertainty about which option has the largest number of dots. In doing so, the modeling of the observed behavior should be expanded in the revised manuscript by describing it and discussing it in the main text, rather than only in the Supplemental Material section.

We thank the reviewer for this very important observation. Indeed, the alternative model suggested by the reviewer is very plausible, and it is paramount to identify the correct source of the bias in sampling. We will now argue that, however, this alternative model does not appear to be driving our results. We will argue this both using different analyses of our data as well as by re-analyzing data from an existing study.

First of all, to rule out the possibility of participants are trying to reduce the uncertainty or internal noise associated with the number of dots presented on screen, we performed a supplementary analysis that included in our regression analysis a predictor that accounts for the level of numerosity for each trial (ΣDots), i.e., the total number of dots presented in the left and right circles. We reasoned that if participants are sampling to reduce the uncertainty generated by numerosity we should see an effect of ΣDots on attention allocation, specifically, a stronger sampling bias towards the chosen item (to compensate for higher uncertainty associated with the higher number of dots presented on display). Therefore, we performed a similar analysis to the one presented in figures 2B and 4B, where ΣDots is included as a predictor of sampling bias instead of confidence. We found that numerosity did not have a significant effect on Sampling Time Difference. We then tested if numerosity of the chosen item (rather than the sum of dots across both stimuli, i.e. ΣDots), affected sampling time, and again we were unable to detect a significant effect, even at a more liberal threshold. (Note that in this analysis for experiment 2 we included only the free condition in which participants controlled information sampling). Both these new analyses are now reported in the supplemental figures.

Furthermore, we also explored whether numerosity influenced participants’ accuracy. We fit a logistic hierarchical regression predicting correct trials and found that in experiment 1 participants made more mistakes in cases of high numerosity (e.g., high ΣDots), but this was a small effect. No significant effect was found in experiment 2. We also performed a hierarchical regression analysis predicting changes of mind, but we again did not find a significant effect of ΣDots (experiment 1: t_1321.78_=0.19, p=0.85; experiment 2: t_287.44_=-0.15, p=0.88).

Despite this small effect on error rate (in experiment 1), we did not find an effect of ΣDots on confidence in any of the experiments. This suggests that participants’ confidence reports were not significantly affected by numerosity, going against the hypothesis that participants were more uncertain (internally noisier) when a high number of dots were on display. These analyses are now included in the supplemental figures.

Also following the reviewers’ advice, we looked more closely at changes in the confidence between the first and second choice. Specifically, we checked whether changes in confidence between choice 1 and 2 were affected by numerosity. We did not find evidence of an effect of ΣDots on confidence change in experiment 2, but we found a significant negative effect in experiment 1 (t = -2.68, p <0.01). This suggests that participants’ confidence changed less between the first and second choice when numerosity was higher. If trials with higher numerosity were perceived as more uncertain/noisy, one would expect larger changes in confidence would be observed in these cases since extra sampling could be especially beneficial to decrease such noise. However, albeit a small effect, the opposite pattern was observed. These analyses are now included in the supplemental figures.

Finally, the reviewers have cleverly suggested a possible additional experiment in which there is also a condition where participants choose the item with the lower number of dots as a way of ruling out the hypothesis that people try to reduce the uncertainty given by high numerosity. While we could not perform this extra experiment, we reanalysed data from a previous perceptual decision study from our group (Experiment 2 in Sepulveda et al., 2020). In this experiment participants had a binary choice of circles with dots with some trials requesting to choose the item with higher numerosity (‘most’ frame), and in others the item with lower numerosity (‘fewest’ frame). This study did not include a resampling stage and second choices, so we checked whether the ‘single’ free sampling stage (eye movements were recorded) was affected by the numerosity of the items. We performed a similar hierarchical linear regression analysis to the one presented above, predicting the difference in sampling time (right – left side was employed in this dataset). We found a significant effect of choice on sampling time difference, which means that participants sampled the item they chose for longer, i.e. the item with highest number of dots in the ‘most’ frame, and the item with lower number of dots in the ‘fewest’ frame. We did not find a significant interaction between choice and ΣDots in this case (if it were instead significant, it would have indicated that overall numerosity in the trial modulated the sampling process of the chosen item). Therefore, if participants’ attention was driven by the need to reduce uncertainty given by numerosity, they should not be gazing at the item with lower number of dots in the ‘fewest’ frame. In short, this new analysis does not support the suggested alternative sampling strategy. This new analysis is now included in the supplemental figures.

Overall, these results seem to indicate that numerosity is not significantly affecting the sampling process. We have included these extra figures in the supplemental figures and the following paragraph in the Results section (line 290):

“… It has been shown that the uncertainty around an internal estimate scales with numerosity (Scott et al., 2015). […] Overall, these results seem to indicate that numerosity is not significantly affecting the sampling process in this task. …”

We have also included the following in the Discussion (line 503):

“… We exclude some alternative explanations for our main findings. Firstly, it could have been possible for participants’ biased sampling to have been driven by a need to reduce uncertainty determined by numerosity (Scott et al., 2015). It is unlikely that this was the case in this task, as neither total numerosity or numerosity of the chosen option significantly predicted sampling bias. …”

2. Task. Some important details concerning the task are missing. In the first phase, it is not clear how the dot patches were delivered to the participants. Figure 1A shows two stimulus presentation phases but the details on the duration of each phase, or whether there was a gap between phases, are missing. In the second phase, the authors should explain more clearly in the task how the sampling phase works, i.e. what the temporal dynamics of responding are. Do subjects press the left key and see the left stimulus until they press right, or does each key press display the stimulus for a period of time? Does the sampling phase start showing one side chosen randomly by the computer, or do subjects need to press a key to start seeing one stimulus? More details about the gaze-contingency and monitoring used in Experiment 2 would help the reader understand how this is done, as would reporting an analysis of the observed looking times for the chosen vs unchosen stimuli in the free vs. fixed sampling conditions (even if only in the Supplementary Material).

The reviewers have rightly suggested we include more details on the task in our manuscript and that we report in more detail actual gaze times in the free and fixed conditions. With respect to the task, we have clarified the timing in Figures 1A and 3A:

We have also altered the Methods section as follows to clarify these points (line 641):

“Experiment 1. The computer task used in this study consisted of 200 trials. On each trial, participants were asked to make binary choices between two dot patches. […] Trials were separated by a 1000ms inter-trial interval (ITI). The task was programmed using Python 2.7.10 and PsychoPy (Peirce, 2007).

Experiment 2. A similar task was used in the second experiment. In this version, in the initial presentation phase each stimulus was presented for 500ms. […] This experiment was programmed using the SR Research Experiment Builder version 1.10.1630 (*SR Research Experiment Builder*, 2017).”

In regards to your request for a more detailed report on actual gaze times in the different conditions, additional details have been included reporting gaze behaviour in Figure 3—figure supplement 1. Note that in this analysis we are using the actual gaze time in the fixed condition rather than the presentation time. An interesting pattern can be observed here: in the fixed condition, participants were inclined to look for longer at the stimulus that was presented for a shorter amount of time. This means that they must have fixated on the covered stimulus while the other stimulus was still being presented (participants’ were only required to fixate inside each circle once during stimulus presentation and could choose to look away) awaiting presentation of the other stimulus. Unfortunately we only have data describing what proportion of the total sampling time participants gazed at each of the two stimuli, but not the exact data describing what proportion of the presentation time of each stimulus participants spent gazing at the stimulus versus waiting for the next stimulus to appear. As such, it is perhaps more difficult to draw conclusions about evidence accumulation in the fixed sampling condition than we previously thought. We have included the following in the main text to reflect this (line 342):

“… However, we also found that on trials where one stimulus was presented longer than the other, participants were inclined to look for longer at the stimulus that was presented for a shorter amount of time (Figure 3—figure supplement 1). This means that they must have chosen to fixate on the covered stimulus while the other stimulus was still being presented awaiting presentation of the other stimulus (participants were only restricted to fixate inside each stimulus once during presentation). In other words, participants often chose to look away when one stimulus was presented for longer than the other in the fixed sampling condition. …”

3. Task. Some aspects of the task deserve further description and discussion. In the first phase, the sequential presentation of the two choice-stimuli invites questions about temporal order biases, both in terms of the initial choice and in terms of the subsequent sampling (how do decision weights change across time). Do people show a choice and a sampling bias toward the first (or last) presented alternative? The confirmation bias effect could occur spuriously if, say, people tend to choose the early alternative and tend to also sample from the early presented alternative due to having forgotten that information (e.g., because of a temporal leak in evidence accumulation). In the second phase, highlighting the chosen alternative with a green arrow could bias the subsequent post-choice sampling towards the chosen alternative (and thus explain part of the observed effect). The authors should either mention this possibility, or state why it is unlikely to drive the observed effect.

We thank the reviewer for pointing out that some aspects of the task were not described and discussed in sufficient detail. We have now rewritten several sections of the manuscript to improve clarity.

First of all, the reviewer rightly points out that presentation order in the first choice phase may bias sampling in the second phase. To rule out this possibility, we performed a hierarchical linear regression analysis predicting Sampling Time Difference (left – right sampling time) using the first presented item (left = 1; right = 0) as a predictor, together with the Difference in Dots (left – right). Since we had available the sequence of item presentation only for our trials in experiment 2, we only show the results for these data. We did not find a significant effect of first item on sampling, in neither the free or fixed condition. Note that, since we only presented each stimulus to the participant once before the first choice phase, we do not include ‘Last item’ as a predictor since it would be correlated with ‘First Item’. For the fixed sampling condition, we are again using the actual gaze time, not the presentation time.

The second part of the reviewer’s comment pertains to sampling phase and the presence of markers on the chosen option. The reviewer correctly points out that this might have interacted with the post-choice sampling phase. We want to clarify that this was a compromise that we reached during the design of the study. Our aim was to minimise memory load, because we were worried that differences in working memory could create a more serious confound in our task. We think it is highly unlikely that these markers can explain away all our effects, since, for example, one would imagine that such low-level visual bias would not be modulated parametrically by factors such as confidence in the first sampling phase and strength of evidence. However, we agree that this should be discussed in the text. In the revised version, we now acknowledge the possibility that these markers may have interacted with sampling and describe the above analysis of presentation order as follows (line 313):

“… The sequential order of presentation in the initial sampling phase before the first choice might also be expected to affect sampling. To exclude this possibility we performed a regression analysis predicting sampling time difference as a function of presentation order in experiment 2 and found no effect (Figure 4—figure supplement 9; t_49.65_=0.08, p=0.93). It’s also important to note that the stimulus chosen in the first choice phase was highlighted throughout the trial. This was done to reduce the likelihood of working memory being a confound on this task, but we recognise the possibility that it may have interacted with the main effect of choice on sampling. …”

4. Contrast between active and passive sampling conditions in Experiment 2. In the passive (fixed) condition, the first choice seems to be less meaningful given that the subjects know that they will subsequently observe more evidence. It would be important to show the psychometric data in Figure 3B separately for active (free) and passive (fixed) sampling and examine if there are any differences between the two conditions (e.g. choice 1 in the fixed sampling condition might be characterised by lower sensitivity/accuracy). Regarding changes-of-mind, is there any evidence that the fixed sampling condition helped counteract potential errors in the first choice, e.g., in cases in which the confirmation-biased free sampling might have led to further committing to an erroneous choice (as suggested in the abstract and the description of the confirmation bias), but the fixed sampling increased the likelihood of changing their mind? Along the same lines, it would be very helpful to know the fraction of changes-of-mind in the two conditions, so that readers can appreciate whether the fixed sampling condition leads to an increase in changes-of-mind. Last, the fact that the effect of sampling bias on changes-of-mind is not there in the fixed sampling condition may not necessarily mean that it is agency itself which drives the effect. Under the authors' hypothesis that subjects aim at sampling from the option with the largest perceived number of dots, the sampling bias measured in the free condition may reflect an endogenous/internal confidence signal regarding the first choice – an internal signal that predicts how likely it is that subjects will confirm this first choice in the second choice. In the fixed condition, this endogenous/internal confidence signal cannot be measured because sampling is controlled by the experimenter, and can only appear as background noise when modeling the second choice. In this view, agency is not driving the effect in a cognitive sense, but only allowing for the effect of an endogenous/internal confidence signal to be detected. This alternative account should either be ruled out by additional analyses or discussed in the main text.

As suggested by the reviewers we included further details on the free and fixed sampling conditions in experiment 2. No differences in performance between the sampling conditions were observed in the psychometric curves for choice 1 (Figure S9.1). However, after participants resampled the items, we observed a divergence in the behaviour in the different conditions. We found that participants had a better performance in identifying the circle with more dots in the free condition (i.e. significantly steeper slope of the psychometric curve; t_17_ = 3.20; p < 0.01). However, we should highlight that overall accuracy in the free condition was not significantly higher than that in the fixed condition in a t-test (Free accuracy: 77.32%, Fixed accuracy: 75.06%; t_17_ = 1.51; p = 0.14).

We did not find evidence indicating that participants were able to correct wrong first decisions more often during fixed condition trials (Figure S9.2A), nor did they have more changes of mind in these trials (Figure S9.2B). So it seems that the fixed condition did not help participants counteract past errors. This observation, together with the finding that there is a slight improvement between the two choices in the free condition, could suggest that information sampled passively is used in a different way to that which participants have chosen to sample themselves. As we mention in the Discussion, this is in line with some other findings on the role of agency in belief updating and decision-making. We have now included this information in the supplemental figures.

Moreover, we have included the following in the Results section (line 354):

“… Furthermore, no significant difference in accuracy was observed between the two conditions (t_17_=1.51, p=0.14), though sensitivity to decision evidence was slightly higher in the second choice in the free sampling condition compared to the fixed sampling condition. The number of changes of mind was also equal between the two conditions (t_17_=0.75, p=0.47) as well as both confidence ratings (confidence in first choice: t_17_=-1.38, p=0.19; confidence in second choice: t_17_=0.5, p=0.62; for more details see Figure 4—figure supplements 10-12). …”

Finally, we have added the following to the Discussion (line 548):

“… It is less clear, however, to what extent the ability to freely sample and any resulting confirmation bias might be beneficial or detrimental to choice accuracy, as in our task there was no clear difference in accuracy between the free and fixed sampling conditions. …”

With respect to the possibility that the sampling bias in the free sampling trials are simply reflecting an internal confidence signal: if this is the case, the observed sampling bias during the free sampling phase should exactly predict the confidence rating on the second choice. We have included some additional analyses of this second confidence rating as a reply to Reviewers’ point #5 (see below), showing that the sampling bias is correlated with confidence in experiment 1, but not at all in experiment 2. So while it is possible that the sampling bias is partly reflecting an endogenous confidence signal, it does not seem likely that it could explain sampling bias entirely. We have added a discussion of this in the main text along with the other analyses of the second confidence rating (see point 5).

5. Analysis of confidence ratings for the second choice. Since confidence ratings were also obtained at the second choice, it seems it would be interesting to analyse those to assess whether these are similarly influenced specifically by actively biased information sampling, e.g. allowing people to feel even more confident in the second choice, but also how that interacts with potential changes of mind. The analysis of confidence ratings for the second choice may also help rule out the possibility that the sampling bias observed in the free sampling condition is driven not by a biased perception of the number of dots toward the previously chosen option, but by an increased internal noise (and larger uncertainty) for higher numerosities (see point 1).

We thank the reviewer for this suggestion. We have now included further analysis of the final confidence in participant’s choices. We already presented some analysis pertaining to the effect of numerosity on confidence (see Reviewers’ point #1, above), which indicates that the overall number of dots did not affect the confidence on the second choice. We performed an additional hierarchical regression analysis specifically predicting the second confidence rating. In both experiments, we found that the second confidence rating was significantly predicted by difficulty (|ΔDots|), with higher confidence in trials with higher dot difference (i.e., easier trials). As expected, we also found that the second confidence rating tended to be lower when participants changed their mind on that trial in both experiments. In experiment 1, we found a positive effect of sampling bias on confidence, indicating that trials in which participants presented more asymmetric resampling towards the chosen option generated higher confidence for the second choice. Interestingly, we found a negative interaction effect between the strength of the sampling bias towards the initially chosen stimulus and change of mind in experiment 1. This means that when participants changed their mind, the effect of sampling bias on confidence was reduced (i.e. gaze asymmetry affected confidence less when participants changed their mind). This may indicate that in trials where participants changed their minds, gaze was less indicative of participants’ actual belief. Importantly, we did not find effects of sampling bias of the chosen item on confidence in the second experiment.

Also, we compared the confidence rating across conditions in experiment 2 and did not find a significant difference.

We have included these results in the Results section as follows (lines 153, 253, and 357):

“… Besides this, sampling time difference in turn affected confidence in the second choice, such that the stronger sampling bias was towards the initially chosen stimulus, the higher confidence in the second choice if that same stimulus was chosen again (Figure 2—figure supplement 1; t_26.01_=9.40, p<0.001). Change of mind was a significant negative predictor of the second confidence rating (t_23.66_=-10.41, p<0.001). Also, a negative interaction effect between sampling time difference and change of mind on the second confidence rating was observed, meaning that the effect of sampling bias on confidence was reversed in trials where participants changed their mind (t_24.48_=-10.21, p<0.001). …”

“… Change of mind also negatively predicted confidence in the second choice phase (Figure 2—figure supplement 1; t_16.59_=-6.39, p<0.001), but in contrast to experiment 1 there was no effect of sampling time difference on the second confidence rating (t_161.30_=-0.78, p=0.44). …”

“… The number of changes of mind was also equal between the two conditions (t_17_=0.75, p=0.47) as well as both confidence ratings (confidence in first choice: t_17_=-1.38, p=0.19; confidence in second choice: t_17_=0.5, p=0.62; for more details see Figure 4—figure supplements 10-12). …”

And the following in the Discussion (line 507):

“… Another possibility is that sampling bias is simply a measure of internal confidence, whereby sampling bias towards the initially chosen stimulus reflects the likelihood of it being chosen again. However, if this were the case a strong relationship between sampling bias and confidence in the second choice would be expected. We only see such a relationship in the first experiment, but not in the second experiment. This suggests sampling bias on this task cannot be fully attributed to an expression of endogenous confidence. …”

6. The illustration of the relation between confidence and sampling bias for the chosen option (Figure 2A and Figure 4A) is confusing. Currently, the curve in Figure 2A seems quite non-monotonic. It is therefore unclear how the conclusion in the title of this figure panel is derived. More precisely, what is confusing is that the correlation stated is an increase in confidence being linked to increased sampling bias, but the data plotted on Figure 2A seem to depict an opposing pattern. This may be because the extreme confidence values show the opposite pattern to intermediate levels, suggesting there isn't a simple linear relation. In fact, the main effect of confidence is negative (in both experiments), which is currently not discussed. What is also unclear is whether this plot is already depicting the choice x confidence interaction? It would be very useful to clarify this illustration, and describe explicitly in the main text whether the model is predicting the sampling time difference between the chosen and unchosen options, or something else (e.g., between the left and right options).

We thank the reviewer for detecting this error. As you pointed out, the plot was showing the difference in sampling (left – right) and not the sampling bias towards the (first) chosen option.

We corrected Figure 4A in the same manner. Note that this figure only shows free sampling trials. We acknowledge that in this case the relationship between sampling bias and confidence may not be as clear, however, the general trend is still positive, as observed from the interaction presented in figure 4B.

This has also been clarified in the Results section as follows (lines 149 and 248):

“… In a hierarchical regression predicting sampling time difference between the chosen and unchosen stimulus, we found a significant interaction between choice and confidence, such that the higher the degree of confidence was in the first choice, the more sampling was biased in favour of that choice (Figure 2; t_26.96_=5.26, p<0.001; see Appendix 1). …”

“… We again found a significant interaction between choice and confidence (Figure 4A-B; t_16.97_=4.29, p<0.001; see Appendix 1) and replicated the main positive effects of choice and evidence difference on sampling time difference between the chosen and unchosen stimulus (Figure 4B; main effect of choice: t_16.97_=2.90, p<0.01; main effect of evidence difference: t_16.97_=9.21, p<0.001). …”

7. Feedback. It is not clear why feedback was not provided on a trial-by-trial basis. More generally, how were participants incentivised in the reported experiment? Did they have reasons to assign equal importance to the first and second choices? More details should be provided regarding these important features of the experiment – given the economic model proposed by the authors makes the assumption that subjects aim at sampling from the option associated with the largest number of dots (a suboptimal strategy) rather than reducing the uncertainty about which option has the largest number of dots (the optimal strategy in the context of the current task).

Participants received a bonus based on their performance at the end of the experiment, even when they did not receive feedback instantly after each trial. Since in this perceptual experiment the objective was quite straightforward and no learning was expected after training, we considered that feedback could insert further confounds. It has been shown that past rewards affect future perceptual decisions in rodents as well as humans (Lak et al., 2020). We have included details on the incentive structure in the Methods section (line 690):

“*Incentive*

The incentive structure used to ensure participants optimized their performance and indicated their true confidence was a linear scoring rule, where the number of points earned on each decision was as follows:

Points=50+correct∗(confidence2)(1)

Here, the variable ‘correct’ is -1 for an incorrect choice and 1 for a correct choice. The variable ‘confidence’ is the confidence rating between 0 and 100. Each of the two choices on a trial was rewarded in this same way, and so should be considered equally important by participants. Participants received an additional £1 for every 3780 points earned on the task. …”

[Editors' note: further revisions were suggested prior to acceptance, as described below.]

You have provided convincing and detailed responses to most comments. The revised manuscript is stronger and provides a more adequate discussion of the findings.Nevertheless, there are two points that we think should be addressed before considering the manuscript suitable for publication at eLife.1. Additional information about the surprising sampling patterns in the fixed condition of Experiment 2. You describe in your response to essential revisions that participants are actually looking at the stimulus presented for a shorter period of time much longer than the stimulus presented for a longer period of time. The graph provided shows that participants look at one stimulus much longer than its presentation time, indicating that participants look inside the blank placeholder while the other stimulus is being presented. This is not central to the paper's findings, but the fixed condition being a control for the free condition, it is important to understand why it may be the case – given the strength and apparent irrationality of the effect. We would like to know how this effect interacts not with chosen/unchosen, but with whether the stimulus presented for shorter was presented first or second. The current explanation for the effect offered in the response letter is currently unclear.

We thank the editors for pointing out this puzzling finding and prompting us to re-examine the data in more depth. To get at the bottom of this issue we decided to rerun our pre-processing steps from scratch using the raw eye-tracking data files. In doing so we found a bug in the pre-processing pipeline we originally used to convert the raw EyeLink data into csv files used for our analyses in Python. This error caused a label switch for the fixed condition causing the surprising pattern of results presented in the previous revisions.

We apologise for this mistake and thank the editors again for highlighting the unusual findings that drove us to find this bug. As you can see from the revised figure 3—figure supplement 1, the pattern of results now make much more sense: patches presented for longer amounts of time are also gazed at longer. This error has not affected any of our findings, but the analyses that included data from the fixed condition have been re-run and as a result some values have slightly changed. We have therefore re-plotted the corresponding figures and have highlighted what has been updated in red in the revised manuscript and supplemental materials.

2. You describe the incentive structure as if it is fully known by the participants. However, it is unclear whether it is indeed the case during the instructions phase. We would like you to clarify whether or not the incentive structure of the task was made explicit to the participants, or whether you only use this incentive structure to derive theory-based interpretations of the observed effects.

The incentive structure was explained to participants in detail during the instruction phase. We have now clarified this in the manuscript (line 713) and by presenting the specific instructions given to participants in the Appendix.

Line 713: “… Here, the variable ‘correct’ is -1 for an incorrect choice and 1 for a correct choice. The variable ‘confidence’ is the confidence rating between 0 and 100. Each of the two choices on a trial was rewarded in this same way, and so should be considered equally important by participants. Participants received an additional £1 for every 3780 points earned on the task. This scoring rule was made explicit to participants (see Appendix 4) …”